# Global carbon budgets estimated from atmospheric $O_2/N_2$ and $CO_2$ observations in the western Pacific region over a 15-year period

Yasunori Tohjima[1], Hitoshi Mukai[1], Toshinobu Machida[1], Yu Hoshina[1], Shin-ichiro Nakaoka[1]

[1]National Institute for Environmental Studies, Tsukuba, 305-8506, Japan

Correspondence to: Yasunori Tohjima (tohjima@nies.go.jp)

**Abstract.** Time series of atmospheric $O_2/N_2$ ratio and $CO_2$ mole fraction of flask samples obtained from NIES's flask sampling network are presented. The network includes two ground sites, Hateruma Island (HAT, 24.05N, 123.81°E) and Cape Ochiishi (COI, 43.17°N, 145.50°E), and cargo ships regularly sailing in the western Pacific. Based on temporal changes in fossil fuel-derived $CO_2$ emissions, global atmospheric $CO_2$ burden, and atmospheric potential oxygen (APO), which were calculated from the observed $O_2/N_2$ ratio and $CO_2$ mole fraction according to $APO=O_2+1.1\times CO_2$, we estimated the global carbon sinks of the ocean and land biosphere for a period of more than 15 years. In this carbon budget calculation, we adopted a correction for the time-varying ocean $O_2$ outgassing effect with an average of 0.54 PgC $yr^{-1}$ for 2000-2016. The outgassing effect, attributed mainly to global ocean warming, was evaluated under the assumption that the net ocean gas flux is proportional to the change in the ocean heat content for the 0-2000 m layer. The resulting oceanic and land biotic carbon sinks were $2.6 \pm 0.7$ PgC $yr^{-1}$ and $1.5 \pm 0.9$ PgC $yr^{-1}$, respectively, for a 17-year period (2000-2016) and $2.4 \pm 0.7$ PgC $yr^{-1}$ and $1.9 \pm 0.9$ PgC $yr^{-1}$, respectively, for a 14-year period (2003-2016). Despite the independent approaches, these sink values of this study agreed with those estimated by the Global Carbon Project (GCP) within a difference of about $\pm 0.4$ PgC $yr^{-1}$. We examined the carbon sinks for an interval of five years to assess the temporal trends. The pentad (5-year) ocean sinks showed an increasing trend at a rate of $0.08 \pm 0.02$ PgC $yr^{-2}$ during 2001-2014, while the pentad land sinks showed an increasing trend at a rate of $0.23 \pm 0.04$ PgC $yr^{-2}$ for 2001-2009 and a decreasing trend at a rate of $-0.22 \pm 0.04$ PgC $yr^{-2}$ during 2009-2014. Although there is good agreement in the trends of the pentad sinks between this study and that of GCP, the increasing rate of the pentad ocean sinks of this study was about two times larger than that of GCP.

## 1. Introduction

In spite of various international efforts to reduce anthropogenic carbon dioxide ($CO_2$) emissions, the atmospheric $CO_2$ levels observed around the world have shown a steady increase and exceeded the benchmark of 400 parts per million mole fraction (ppm) in past years (Betts et al., 2016). The Carbon Dioxide Information Analysis Center (CDIAC) reported that global fossil fuel-derived $CO_2$ emissions in recent years still increased gradually and rose to 9.9 PgC $yr^{-1}$ by 2014 (Boden et al. 2017). Under these circumstances, the Paris Agreement adopted at COP21 in 2015 aimed to reduce the anthropogenic greenhouse gas emissions to maintain the increase in global mean surface temperatures well below 2°C by 2100, and if possible, to limit the increase to 1.5°C. To achieve this goal, it is crucially important to quantitatively understand the natural sink strengths or land biosphere and ocean sinks. A variety of approaches have so far been applied to the quantification of ocean or land sinks or both, including process-based land and ocean models, bottom-up emission estimates based on flux measurements, and top-down estimates based on atmospheric measurements. Developing process-based models to enhance the accuracy of the global carbon budget is crucially important because they are expected to predict the future global carbon cycle in a warmer world. However, carbon budget estimates based on observations are still important to validate and improve the process-based models.

The budget estimation based on atmospheric $CO_2$ and $O_2$ observations is a simple and straightforward approach, and it has historically settled the controversy whether the land biosphere is a net carbon sink or source (Keeling and Shertz, 1992). Although several techniques based on interferometer (Keeling, 1988), mass spectrometry (Bender et al., 1994), paramagnetic analyzer (Manning et al., 1999), fuel cell analyzer (Stephens et al., 2007), vacuum ultraviolet absorption photometer (Stephens et al, 2003) and so on, have been developed to detect the ppm level changes in the atmospheric $O_2$ concentration, the accurate quantification of the $O_2$ change is still challenging. The carbon budget is evaluated by simultaneously solving the mass balance equations of the atmospheric $CO_2$ and $O_2$ as follows (Manning and Keeling, 2006):

$$\Delta CO_2 = F - B - O, \qquad (1)$$

$$\Delta O_2 = -\alpha_f F + \alpha_B B + Z, \qquad (2)$$

where $\Delta CO_2$ and $\Delta O_2$ represent the changes in the atmospheric $CO_2$ and $O_2$ burdens based on atmospheric observations, respectively, $F$ represents the fossil fuel-derived emissions, and $B$ and $O$ represent the uptake by the land biosphere and the ocean, respectively. $\alpha_F$ and $\alpha_B$ are the $-O_2/C$ exchange ratio for the globally averaged fossil fuel combustions and land biotic processes, respectively. The estimated value for $\alpha_B$ is about 1.1 (Severinghaus, 1995) and that for $\alpha_F$ is about 1.4 (Keeling, 1988). These equations mean that the $CO_2$ and $O_2$ fluxes associated with fossil fuel combustion and land biotic processes are tightly coupled. In contrast, the ocean $CO_2$ uptake, $O$, and ocean $O_2$ emissions, denoted as $Z$, are decoupled because the ocean acts as a carbon sink by physicochemically dissolving the $CO_2$. Since the values of $F$ and $\alpha_F$ can be evaluated from energy statistics (Keeling, 1988), we can evaluate the ocean and land uptake by solving the above equations if we could evaluate the value of $Z$.

The global carbon budget can also be related to tracer atmospheric potential oxygen (APO), which is defined by the equation

of APO=$O_2+\alpha_B \times CO_2$ (Stephens et al., 1998). Since the APO is defined to be invariant with respect to the land biotic exchange, the secular trend in the APO is determined by fossil fuel combustions which cause a gradually decreasing trend in APO, and the air-sea gas exchange. Combining Eq. (1) multiplied by $\alpha_B$ and Eq. (2) in accordance with the APO definition, results in the following equation for APO budget (Manning and Keeling, 2006):

$$\Delta APO = -(\alpha_F - \alpha_B)F - \alpha_B O + Z. \qquad (3)$$

Since observation sites for atmospheric $O_2$ are still limited compared with those for atmospheric $CO_2$, Manning and Keeling (2006) proposed an alternative approach that the global carbon budgets could be obtained by simultaneously solving Eqs. (1) and (3) and using globally averaged $CO_2$ data based on NOAA/ESRL/GMD's measurements. This approach, making maximum use of the available data, is expected to give the most reliable estimation. This APO approach has been adopted for
the estimation of global carbon budget based on atmospheric $O_2$ and $CO_2$ measurements (e.g. Manning and Keeling, 2006; Tohjima et al., 2008; Ishidoya et al., 2012; Goto et al., 2017).

To evaluate the carbon budgets based on $O_2$ and $CO_2$ measurements, we need to quantify the magnitude of $Z$ and its temporal variation, if possible. It is considered that $Z$ has a large interannual variability because observed trends of APO generally show large interannual variations which would result in unrealistic variations in the ocean uptake if the variability in $Z$ were rather
small (e.g. Bender et al., 2005). Probably, an imbalance of the air-sea seasonal $O_2$ exchanges, outgassing flux associated with primary production in spring and summer and ingassing flux associated with ocean ventilation in autumn and winter, cause the interannual variations in $Z$. The results of ocean model simulations also support this mechanism (e.g. McKinley et al., 2003; Nevison et al. 2008). Therefore, it is difficult to estimate the short-term carbon budgets unless the temporal variations in $Z$ are accurately evaluated. Additionally, as for long timescales, it is considered that the present ocean acts as an $O_2$ source
because of the global ocean warming (Keeling and Garcia, 2002). The increase in surface ocean temperature not only reduces the solubility of gases in seawater but also strengthens the ocean stratification, which reduces the ventilation of interior water masses. The reduction of ventilation reduces the ingassing flux of $O_2$. In contrast, the reduction of ventilation also causes a reduction of the nutrient supply from deep water, which might decrease the primary production and $O_2$ outgassing in summer. Therefore, the influence of the ocean warming on the net air-sea gas exchange is rather complicated.

Unfortunately, there is little observational evidence to quantify the magnitude of $Z$ and its temporal variations. The long-term average values of $Z$ are inferred under the assumption that $Z$ is proportional to the air-to-sea heat flux (Keeling and Garcia, 2002). The change in the global ocean heat content has been evaluated based on the large data set of ocean observations (Levitus et al. 2012). Keeling and Garcia (2002) estimated the $O_2$ flux / heat flux ratio from the relationship between the dissolved $O_2$ corrected for the mineralization effect and the potential temperature. This approach was basically adopted by
most of the studies to evaluate the long-term global carbon budgets (e.g. Bender et al., 2005; Manning and Keeling, 2006; Tohjima et al., 2008). On the other hand, Ishidoya et al. (2012) evaluated the instantaneous variations in the land and ocean sinks based on the APO data at Ny-Ålesund, Svalbard, and Syowa, Antarctica for the period 2001-2009 by calculating the

interannual variation in Z from the temporal variation in the ocean heat content. They concluded that the above-mentioned $Z$ values adequately suppressed artifacts caused by the imbalance of the seasonal air-sea $O_2$ exchange.

We have been conducting air sampling into glass flasks for the measurement of the atmospheric $O_2/N_2$ ratio and $CO_2$ mole fraction at two ground sites in Japan since the late 1990s (Tohjima et al., 2003), and have been evaluating the global carbon budgets for up to 7 years (1999-2005) based on the APO data from the flask observations (Tohjima et al., 2008). To extend the observation area, we started additional flask sampling aboard commercial cargo ships regularly sailing in the Pacific region in 2002 (Tohjima et al., 2005b, 2012). About a decade has passed since we previously reported the global carbon budgets, and now we have more than a 15-year long record of atmospheric $O_2/N_2$ and $CO_2$ of the flask samples. In this study, we estimated the ocean and land biotic carbon sinks for over a decade by using the temporal changes in the APO based on these flask data. In addition, we sequentially computed the ocean and land sinks for an interval of five years and examined the changing trends of both sinks. In these budget calculations, we estimated the values of $Z$ for the corresponding period by using the temporal changes in the global ocean heat content. Finally, the estimated ocean and land carbon sinks of this study were compared with those of the Global Carbon Project (GCP).

## 2. Data and analysis

### 2. 1. Flask sampling locations

We started air samplings for the measurement of the atmospheric $O_2/N_2$ ratio and $CO_2$ mole fraction at two monitoring stations located on Hateruma Island (HAT, 24.05N, 123.81°E) in July 1997 and at Cape Ochiishi (COI, 43.17°N, 145.50°E) in December 1998 (Tohjima et al., 2003). In addition, we have been collecting air samples from the Pacific regions by using commercial cargo vessels equipped with automated flask sampling systems (Tohjima et al., 2005b; 2012). The shipboard flask samplings were started between Japan and North America in December 2001, between Japan and Australia/New Zealand in December 2001, and between Japan and Southeast Asia in September 2007. The flask sampling sites are depicted in Figure 1. Unfortunately, the shipboard data in Southeast Asia, the northern North Pacific (north of 30°N), and the eastern North Pacific are spatiotemporally rather sporadic. (See the inserted figure in Fig. 1 showing time-latitude plots of the shipboard flask samples.) Thus, in the following analysis, we only used the data set obtained at HAT, COI and the western Pacific region between 40°S and 30°N and between 130°E and 180°E.

Air samples were collected in glass flasks hermetically sealed by two glass valves with Viton O-rings. The volumes of the flasks were 2 liters for the samplings at HAT and COI and 2.5 liters for the shipboard samplings. It should be noted that glass flasks with a volume of 1 liter were also used in the early period from the start to March 2006 and only 1-liter flasks were used from the start to January 1999 at HAT.

### 2. 2. $O_2/N_2$ and $CO_2$ analytical methods

In this study, we used a gas chromatograph (GC) equipped with a thermal conductivity detector (TCD) for the measurements of atmospheric $O_2$ (Tohjima 2000). In this GC/TCD method, $O_2/N_2$ ratios of sample air and working reference air were alternately measured and the atmospheric $O_2$ change was determined as the relative difference in the $O_2/N_2$ ratio from an arbitrary reference. We used the delta notation according to Keeling and Shertz (1992) to express the relatively small difference in the $O_2/N_2$ ratio as follows:

$$\delta(O_2/N_2) = \frac{(O_2/N_2)_{sam}}{(O_2/N_2)_{ref}} - 1, \qquad (4)$$

where subscripts "*sam*" and "*ref*" refer to sample and reference, respectively, and the $\delta(O_2/N_2)$ value multiplied by $10^6$ is expressed in "per meg" units. The change in 1 μmol of $O_2$ per mole of dry air changed the $O_2/N_2$ ratio by 4.77 per meg, which corresponds to 1 ppm change in the atmospheric trace gas abundance. APO was calculated from the $CO_2$ mole fraction ($X_{CO2}$) in ppm and $\delta(O_2/N_2)$ in per meg according to

$$\delta APO = \delta(O_2/N_2) + \alpha_B X_{CO_2}/S_{O_2} - 1850, \qquad (5)$$

where $S_{O2}$ is the mole fraction of $O_2$ in the air ($S_{O2}$=0.2094, Tohjima et al., 2005a) and the value of 1850 is an arbitrary reference point of $\delta APO$ in per meg. The values of $\delta(O_2/N_2)$ were determined against the NIES $O_2/N_2$ scale (Tohjima et al. 2008). Its temporal stability is examined in the following section.

A nondispersive infrared (NDIR) analyzer (LI-COR, Lincoln, Ne., model LI-6252) was used for the $CO_2$ measurement of the flask samples. The $CO_2$ mole fractions were determined against the NIES 09 scale, which is based on a set of gravitationally prepared $CO_2$-in-air standard gases (Machida et al., 2011). The relationship between the NIES 09 scale and the NOAA scale were repeatedly compared through the WMO Round-robin inter-comparison program. The results showed that the differences of the NIES 09 scale from the NOAA scale were kept within ±0.15 ppm during the period from 1996 to 2014 (https://www.esrl.noaa.gov/gmd/ccgg/wmorr/wmorr_results.php?).

## 2. 3. $O_2/N_2$ scale stability

As details of the NIES $O_2/N_2$ scale are given elsewhere (Tohjima et al., 2008), here we describe only briefly the outline of the scale and add some new information below. The zero point of NIES $O_2/N_2$ scale had been related to an ambient dry air stored in a high-pressure cylinder (HDA-1). The $O_2/N_2$ scale was maintained by three cylinders during the first four years (1997-2001) of our $O_2/N_2$ measurement program. In 2001, the NIES $O_2/N_2$ scale was transferred to 11 other high-pressure cylinders (five 10-liter cylinders and six 48-liter cylinders), of which the $\delta(O_2/N_2)$ values were carefully determined against the original $O_2/N_2$ scale. Another primary reference gas (48-liter cylinder, CQB-07080) was added in 2002, and now 12 primary reference gases keep the NIES $O_2/N_2$ scale. The air samples delivered from glass flasks or high-pressure cylinders were measured against the working reference airs stored in 48-liter aluminum cylinders, of which the $\delta(O_2/N_2)$ values were repeatedly determined against the individual primary gas cylinders at intervals of a few months. The working reference gas cylinders were replaced

by new ones every one to two years.

Fig. 2 is the extended version of the previously reported figure (Fig. 1 in Tohjima et al., 2008), showing the temporal changes in the $O_2/N_2$ ratio of primary reference gases relative to the NIES $O_2/N_2$ scale. In the figure, the deviations of the $O_2/N_2$ ratio from the average value for HDA-2 and from the initially determined values for the second set of 12 cylinders are plotted. The averages and the standard deviations (1σ) of the differences for the individual 12 cylinders range from −4.2 per meg to 3.3 per meg and from 3.1 per meg to 5.2 per meg, respectively. The changing rates of the deviations for the 12 reference gases during 2001-2017, determined by least square linear regression, range from −0.34 per meg yr$^{-1}$ to 0.2 per meg yr$^{-1}$. Solid and broken horizontal bars in the bottom of the figure indicate the durations of use of the individual working reference gases.

To assess the stability of the NIES $O_2/N_2$ scales, we have continued to measure the reference gases in two 48-L aluminum cylinders (CQB-15645 and CQB-15649) since 2003, which are independent from the reference gases for the NIES scale. The results are shown in Figure 3. The average changing rates for the whole period, evaluated by linear regression analysis, are −0.14 ± 0.06 per meg yr$^{-1}$ for CQB-15645 and −0.05 ± 0.06 per meg yr$^{-1}$ for CQB-15649. Therefore, we conclude that the stability of the NIES $O_2/N_2$ scale has been maintained within ±0.2 per meg yr$^{-1}$ at least during the period of 2003-2016. However, this stability test cannot exclude the possibility that the $O_2/N_2$ ratios of the reference gases drift across all the cylinders rather uniformly. There are several mechanisms that affect the $O_2/N_2$ ratios of the gases within the high-pressure cylinders, including corrosion of the inner surface, leakage, thermal diffusion and gravitational fractionation. Keeling et al. (2007) assessed carefully and comprehensively the influences of those potential mechanisms on the long-term stability of the $O_2/N_2$ ratio of the reference gases and obtained an estimated uncertainty of ±0.4 per meg yr$^{-1}$. We also treated the reference cylinders, which were kept horizontally in a thermally insulated box, with the greatest care (Tohjima et al., 2008). Therefore, we adopted the value of ±0.4 per meg yr$^{-1}$ as the long-term drift of the reference gases caused by the above degradation effects. Consequently, we assumed that the total uncertainty of the long-term stability of the $O_2/N_2$ reference scale was ±0.45 per meg yr$^{-1}$ (=$(0.2^2+0.4^2)^{1/2}$) in this study.

## 2. 4. Carbon budget calculation

The ocean and land uptake, $O$ and $B$, are given by the following equations (Manning and Keeling, 2006; Tohjima et al., 2008):

$$O = \left[-(\alpha_F - \alpha_B)F - \left(\frac{S_{O_2}}{\beta}\right) \times \Delta APO + Z_{eff}\right] \times \frac{1}{\alpha_B}, \qquad (6)$$

$$B = \left[\alpha_F F + \left(\frac{S_{O_2}}{\beta}\right) \times \Delta APO - \left(\frac{\alpha_B}{\beta}\right) \times \Delta X_{CO_2} - Z_{eff}\right] \times \frac{1}{\alpha_B}, \qquad (7)$$

where $\beta$ is the coefficient converting PgC to ppm $CO_2$ in the atmosphere ($\beta$=0.470 ppm/PgC, Tohjima et al., 2008), and $Z_{eff}$ represents the net effect of the air-sea $O_2$ and $N_2$ exchange on the atmospheric $O_2/N_2$ ratio. In these equations, $O$, $B$, $F$ and $Z_{eff}$ are given in units of PgC yr$^{-1}$, $\Delta APO$ in units of per meg yr$^{-1}$, and $\Delta X_{CO2}$ in units of ppm yr$^{-1}$. Note that $F$ and $\alpha_F$ include the $CO_2$ emissions associated with cement manufacturing. The values of $\alpha_F$ were calculated from the $CO_2$ emission amounts and

the $-O_2/CO_2$ molar exchange ratios of the individual fuel types (Keeling 1988). Since $\alpha_F$ slightly varies year by year, the value of $\alpha_F$ for the relevant period was used for the carbon budget calculations. The values of $Z_{eff}$ were calculated based on the effects of global ocean warming and anthropogenic nitrogen deposition in accordance with the approach of Keeling and Manning (2014). The details of the $Z_{eff}$ calculation is discussed in the following section.

We used the same data set of the fossil fuel-derived $CO_2$ emissions and the global average of the atmospheric $CO_2$ mole fractions as the Global Carbon Project (GCP) used for the estimation of global carbon budget 2018 (Le Quéré et al. 2018). The fossil $CO_2$ emissions from fossil fuel combustion and cement production were basically based on the dataset from CDIAC and other energy statistics (Boden et al, 2017). The change in the atmospheric $CO_2$ burden was calculated based on the global observation by the US National Oceanic and Atmospheric Administration Earth System Research Laboratory (NOAA/ESRL; 

Dlugokencky and Tans, 2018). The temporal variations in the fossil $CO_2$ emissions and the atmospheric $CO_2$ accumulation rate are depicted in Fig. 4.

Annual means of APO centered on January 1 were computed by using the same procedure as Tohjima et al. (2008). First, smooth curve fits to the data were computed in accordance with the methods of Thoning et al. (1998) with a cut-off frequency of 4.6 cycles $yr^{-1}$. Then the flask APO data were modified to represent the values at the center of the individual months by 

shifting them in parallel with the smooth curve fits. This procedure aimed to reduce the influence from biases of the sampling timings within the individual months. The monthly averages were calculated from the modified APO data. When there were no flask data in the monthly time frame, the monthly average of the smooth curve was used. The annual means were calculated from the consecutive 12 monthly averages from July to June of the following year. The standard errors of the differences between the flask data and the smooth curve fits for the corresponding annual periods were adopted as the uncertainties for the 

annual averages. The averages and ranges (minimum ~ maximum) of the errors for the annual means of APO were 0.8 per meg (0.6~1.3 per meg) for HAT and 1.1 per meg (0.9 ~ 1.4 per meg) for COI.

In this study, we adopted the value of 1.10 of Severinghaus (1995) for $\alpha_B$ in accordance with a series of previous studies (e.g. Bender et al., 2005; Manning and Keeling, 2006; Ishidoya et al., 2012; Keeling and Manning, 2014; Goto et al., 2017). However, several studies (e.g. Randerson et al., 2006; Worrall et al., 2013), investigating the elemental compositions of organic matters 

in soil and plants, indicated that the value of 1.1 is rather large for the globally averaged net $-O_2/CO_2$ exchange ratio for the terrestrial biosphere. These studies suggest that the value of 1.05 is much more appropriate for $\alpha_B$. Although it's beyond the scope of this study to discuss which value is better for $\alpha_B$, it is useful to mention that the use of 1.05 for $\alpha_B$ results in larger decreasing rates of APO by about 5% and an increase in land sinks and a decrease in ocean sinks by about 0.06 PgC $yr^{-1}$ on average in the following results. Considering recent reports about the global net $-O_2/CO_2$ exchange ratio, Keeling and Manning 

(2014) revised the uncertainty of $\alpha_B$ upward from $\pm0.05$ (Severinghaus, 1995) to $\pm0.10$. Thus, we also adopted $\pm0.10$ for the uncertainty of $\alpha_B$ in this study.

## 2. 5. Evaluation of outgassing effect ($Z_{eff}$)

As is discussed in the Introduction, today's ocean is considered to act as a net source of atmospheric $O_2$ because of global ocean warming, which also affects the air-sea $N_2$ exchange. Since the atmospheric $O_2$ change is measured as the change in the atmospheric $O_2/N_2$ ratio, the outgassing effect caused by the global ocean warming, which is denoted as $Z_{gow}$, should include the influences from not only ocean $O_2$ outgassing but also ocean $N_2$ outgassing. Assuming the relationship is proportional

between the gas fluxes and heat fluxes across the air-sea interface, Manning and Keeling (2006) gave the equation for $Z_{gow}$ as:

$$Z_{gow} = \left( \gamma_{O_2} - \frac{S_{O_2}}{S_{N_2}} \gamma_{N_2} \right) \times Q \times m_c \times 10^{-15}, \qquad (8)$$

where $Q$ represents the changing rate of the global ocean heat storage in units of J yr$^{-1}$, $\gamma_{O2}$ and $\gamma_{N2}$ are gas flux / heat flux ratios between the air and sea in units of mol J$^{-1}$, $S_{N2}$ is the mole fraction of atmospheric nitrogen ($S_{N2}$=0.7809, Tohjima et al., 2005b) and $m_c$ is the atomic mass of carbon ($m_c$=12.01). $Z_{gow}$ is given in units of PgC yr$^{-1}$.

The primary mechanism that affects the air-sea gas exchange is a reduction of gas solubility caused by the increase in the ocean temperature. Therefore, the gas flux / heat flux ratio derived from the above thermal effect can be evaluated from the temperature dependence of gas solubility in the seawater and the specific heat of the seawater. Since the air-sea $N_2$ exchange is predominantly driven by the thermal effect, we adopted the estimated $\gamma_{N2}$ of 2.2 nmol J$^{-1}$ in this study in accordance with previous studies (Keeling and Garcia, 2002; Manning and Keeling, 2006).

In contrast to the air-sea $N_2$ exchange, the changes in the ocean circulation and ocean primary production also affect the air-sea $O_2$ exchange as is mentioned in the Introduction. Examining the ratio of the seasonal ocean outgassing of $O_2$ to the seasonal ocean heating and the negative linear relationship between the dissolved $O_2$ concentrations corrected for ocean biological processes and the potential temperature in the main thermocline based on archived global observation data, Keeling and Garcia (2002) obtained the estimate of 4.9 nmol J$^{-1}$ for $\gamma_{O2}$. The value of $\gamma_{O2}$ was also investigated by using ocean biogeochemical

models to revise the global carbon budgets based on $O_2$ observations (e.g. Plattner et al., 2002; Bopp et al., 2002). Keeling et al. (2010) summarized the model-based values of $\gamma_{O2}$ ranging from 5.9 to 6.7 nmol J$^{-1}$. On the other hand, Stendardo and Gruber (2012) examined a huge archived dataset of observations in the North Antarctic Ocean during the past five decades and obtained changing ratios of $O_2$ inventory to heat content of $-4.3 \pm 2.4$ nmol J$^{-1}$ in the upper 700m and $-1.6 \pm 1.9$ nmol J$^{-1}$ between 700 and 2750 m. These basin-scale ocean $O_2$/heat changing ratios seem to suggest that the global ocean acts as a net

$O_2$ source due to global ocean warming. Therefore, in this study, we used the value of 4.9 nmol J$^{-1}$ for $\gamma_{O2}$ according to previous studies.

To compute Q, we used the estimates of the world ocean heat content (OHC) based on a variety of oceanographic data (Levitus et al., 2000; Levitus et al., 2012). Time series of OHC for the 0-700 m and 0-2000 m layers are available from the NOAA's web site (https://www.nodc.noaa.gov/OC5/3M_HEAT_CONTENT/). In previous carbon budget estimations based on

atmospheric $O_2/N_2$ measurements, the values of Q were estimated from the OHC for the 0-700 m layer (e.g. Manning and Keeling, 2006; Tohjima et al., 2008; Ishidoya et al. 2012). Levitus et at. (2012) showed, however, that the ocean heat storage

of the 700-2000 m layer contributes to about one third of the total heat storage of the 0-2000 m layer. Keeling and Manning (2014) estimated the value of Q by considering not only Q for the depths above 700 m but also Q for the depths below 700, which contributed to 30% of the total value of Q. Therefore, the time series of OHC for the 0-2000 m layer was used in this study. Note that since the annual average of OHC for the 0-2000 m layer is available only after 2005, we used the pentad (5-year) averages before 2005.

In addition to the ocean warming effect, Keeling and Manning (2014) introduced recently another ocean outgassing effect caused by atmospheric deposition of excess anthropogenic nitrogen to the open ocean. The excess nitrogen is considered to enhance the ocean biotic production of organic matter, which is associated with the $O_2$ production. Keeling and Manning (2014) evaluated the anthropogenic nitrogen-induced outgassing as being about $0.1 \times 10^{14}$ mol $O_2$ yr$^{-1}$. Since the outgassing effect caused by the anthropogenic nitrogen deposition is small but rather significant, we adopted the effect as $Z_{anthN}$, with a magnitude of 0.12 PgC yr$^{-1}$ (=$0.1 \times 10^{14}$ mol $O_2$ yr$^{-1}$ ×12.01 gC mol$^{-1}$). Eventually, the total outgassing effect, $Z_{eff}$, is expressed as the summation of $Z_{gow}$ and $Z_{anthN}$:

$$Z_{eff} = Z_{gow} + Z_{anthN}. \qquad (9)$$

The time series of the annual $Z_{eff}$ divided by $\alpha_B$ are depicted in Fig. 4 as purple lines. The value of $Z_{eff}/\alpha_B$ ranges from 0.2 Pg yr$^{-1}$ to 1.0 Pg yr$^{-1}$, and the 19-year average for 1998-2016 is 0.52 Pg yr$^{-1}$. There are not much differences in $Z_{eff}$ between this study and previous studies (e.g. Manning and Keeling, 2006; Keeling and Manning, 2014; Tohjima et al., 2008; Ishidoya et al., 2012; Goto et al., 2017).

## 3. Results and discussion

### 3. 1. Observed $O_2/N_2$, $CO_2$, and APO

The time series of the atmospheric $CO_2$ mole fraction, $O_2/N_2$ ratio, and APO of the air samples collected at HAT and COI and onboard cargo ships sailing between 40°S and 30°N in the western Pacific are depicted in Fig. 5 together with the smooth-curve fits. The ship data were binned into 10-degree latitudinal bands (40-30°S, 30-20°S, …, 20-30°N). Note that there are no data gaps with more than 50 days in the time series at HAT and COI while the time series of TF5 have data gaps during the 7-month period from October 2006 to April 2007. The ship data during the 7-month period were significantly contaminated by the inboard air due to the failure of diaphragm of the sampling pump.

The temporal variations in the annual means of the atmospheric $CO_2$, $O_2/N_2$ and APO, centered on January 1st, are shown in Figs. 6a, 6b, and 6c, respectively, where linear trends obtained from least square fitting to the data of HAT were subtracted from the individual time series to emphasize the interannual variations. The standard errors of the annual means for HAT and 10-0°S bin are depicted as vertical bars for typical examples. Note that the annual means of the atmospheric $CO_2$, $O_2/N_2$ and APO and the corresponding standard errors for HAT, COI and the 10-degree bins are summarized in Table S1-S9 in the

Supplement. The annual means of $CO_2$ and $O_2/N_2$ show latitudinal gradients of northward increase and southward increase, respectively, because the fossil fuel-derived $CO_2$ emissions and $O_2$ consumptions occur predominantly in the northern mid-latitudes. In contrast, the highest values of the annual mean APO were generally observed around the equator as previously reported (Battle et al., 2006; Tohjima et al., 2005b, 2012). This equatorial peak is mainly attributed to large-scale air-sea gas

exchanges: ingassing in the mid- and high-latitudes and outgassing in the equatorial region. Conducting atmospheric inversion analyses based on the APO data from the Scripps observation network, Rödenbeck et al. (2008) suggested anomalous outgassing of APO from the equatorial region during El Niño periods, while Tohjima et al. (2015) found a suppressed equatorial peak during El Niño periods based on the western Pacific observations. Eddebbar et al. (2017) reconciled these conflicting results by predicting the existence of a zonal dipole-like ENSO response in the equatorial Pacific based on several ocean

process-based models and an atmospheric transport model. These results suggest that an enhanced zonal coverage of the atmospheric observations in the equatorial Pacific is needed to constrain the full basin-scale ENSO response. We can see a considerable suppression of the equatorial peak during the strong 2015/2016 El Niño event in Fig. 6c, which was not reported in Tohjima et al. (2015). Any detailed discussion about the temporal variation of the equatorial peak during the 2015/2016 El Niño event is, however, beyond the scope of this study and will be given elsewhere.

Fig. 6d shows the time series of the annual changes in the annual mean APO, which are the annual changing rates of APO for a one-year interval ($\Delta t$ =1 year). As you can see, there are considerable differences in the annual changing rates among the observation sites in the same years; the standard deviations range from 1.6 to 4.4 per meg and the average is 2.8 per meg. We also depict the averages of the annual changing rates of APO of HAT and COI and of all the shipboard data as thick grey line in Fig. 6d. Note that these average annual changing rates of APO were used for calculation of the global carbon budget in the

following sections. The average annual changing rates show also a large interannual variability with a standard deviation of 4.7 per meg $yr^{-1}$ for the entire observation period.

The differences in the changing rate of APO among the sites in the same years decrease with increase in the interval for the calculation ($\Delta t$) as shown in Fig. 7, where the average (red circles) and the minimum and maximum (red broken lines) standard deviations of the changing rates are plotted against $\Delta t$. The differences among the sites decrease almost inversely with $\Delta t$; the

average standard deviation for $\Delta t$=5 year is 0.54 per meg $yr^{-1}$. The temporal variability in the changing rate also decreases inversely with $\Delta t$ as depicted in Fig. 7 (blue circles); the standard deviation is reduced to 1.2 per meg $yr^{-1}$ for $\Delta t$ =5 years. The above results seem to suggest that the temporal variability in the APO fluxes exceeds the spatial variability. As is indicated by Eq. (3), the temporal variability in the APO changing rate should be attributed mostly to those in $O$ and $Z_{eff}$. Therefore, the above results also indicate that an interval of 5 years could suppress the temporal variability in $Z_{eff}$ to the level of $\pm1.2$ per meg

$yr^{-1}$, which corresponds to a carbon budget of about $\pm0.5$ PgC $yr^{-1}$.

The changing rates of the atmospheric $CO_2$, $O_2/N_2$, and APO for several combinations of time periods and the observed data (HAT, COI and shipboard) are summarized in Table 1. Here, the uncertainties of the changing rates were computed from the uncertainties of the corresponding annual means at both ends of the periods, the estimated uncertainty of the $O_2/N_2$ scale

stability ($\pm 0.45$ per meg yr$^{-1}$, Section 2. 3), and the uncertainty in the $O_2/N_2$ span sensitivity ($\pm 3\%$, see below). The time periods of 2000-2010, 2001-2010, and 2001-2014 were selected to compare the observational results of this study with those of Scripps Institution of Oceanography (SIO) and Tohoku University (TU) (Keeling and Manning, 2014; Ishidoya et al., 2012; Goto et al., 2017). As is discussed in the above section, the differences in the long-term changing rates of APO between HAT, COI and shipboard data are less than 0.3 per meg yr$^{-1}$, while the increasing rates of $CO_2$ and the decreasing rates of $O_2/N_2$ for HAT are slightly larger than those for COI and other sites. The monitoring station of HAT is located at the marginal region of continental East Asia, and the anthropogenic $CO_2$ emissions from China often influence the observations at HAT during winter due to the East Asian monsoon (c.f. Minejima et al., 2012; Tohjima et al, 2010, 2014). Additionally, for the period of 2000-2014, the fossil fuel-derived $CO_2$ emissions from China show a rapid increase in association with the unprecedented economic growth. These situations may explain the rather large increase in $CO_2$ and decrease in $O_2/N_2$ at HAT. In contrast to $CO_2$ and $O_2/N_2$, the emissions from fossil fuel combustion and land biotic processes contribute less to the APO variations, resulting in relatively small differences in the long-term APO changing rates among the sites.

It should be noted that the decreasing rates of APO of our study are $0.5 \sim 1.1$ per meg yr$^{-1}$ smaller than those of SIO and TU. Except for the differences of the observation sites, we can offer two explanations for the discrepancy. First, the calculation methods of the changing rate adopted by Goto et al. (2017) are different from those adopted in this study, which might partially explain the discrepancy. This is understandable when comparing the changing rates of $CO_2$, $O_2/N_2$, and APO for the individual studies. In this study, the APO changing rates are almost consistent with those calculated from the $CO_2$ and $O_2/N_2$ changing rates according to the APO definition. However, in the study of Goto et al. (2017), the $CO_2$ and $O_2/N_2$ changing rates of Ny-Ålesund give APO decreasing rates of $-9.4$ per meg yr$^{-1}$, which is 0.7 per meg yr$^{-1}$ smaller than the originally reported values. Second, inter-laboratory comparison of flask samples and high-pressure cylinders suggests a possibility that the span sensitivity of the $O_2/N_2$ measurements of NIES is about 3% lower than that of SIO, which can almost explain the differences in the APO decreasing rates. However, to obtain an accurate conclusion, we need much more studies.

### 3. 2. Calculation of global carbon budgets

The rates of the global carbon uptake by the ocean and land biosphere were calculated from the average changing rate of APO based on observations at COI, HAT and on cargo ships (40ºS-30ºN). The results for several time periods are summarized in Table 2 together with the average changing rates of APO, globally averaged atmospheric $CO_2$ accumulation rates, fossil fuel-derived $CO_2$ emission rates, and the ocean outgassing effect divided by $\alpha_B$. The 19-year (1998-2016), 17-year (2000-2016), and 14-year (2003-2016) periods correspond to the individual maximum observation periods for HAT, COI, and the western Pacific, respectively. For example, the estimated ocean and land sinks for 2000-2016 were found to be $2.6\pm0.7$ PgC yr$^{-1}$ and $1.5\pm0.9$ PgC yr$^{-1}$, respectively.

The uncertainties in the parameters used for the carbon budget calculation (Eqs. (6) and (7)), which are also listed in Table 2, are briefly discussed here. Note that in this study the estimated uncertainties are $\pm 1\sigma$. Since the ocean outgassing effect is rather

speculative, we assumed that the values of $Z_{eff}$ for the individual periods had ±100% uncertainties in accordance with previous studies (e.g. Manning and Keeling, 2006; Tohjima et al., 2008). We adopted uncertainties of ±5% for the fossil fuel-derived $CO_2$ emission rate and ±0.2 PgC yr$^{-1}$ for the atmospheric $CO_2$ increasing rate from Le Quéré, et al. (2018). As for the uncertainties of the observed APO changing rates, we adopted the standard deviations among the sites shown in Fig. 7 (±0.37

per meg yr$^{-1}$ for longer than 10 years and ±0.54 per meg yr$^{-1}$ for 5 years). The estimated uncertainty of the $O_2/N_2$ scale stability (±0.45 per meg yr$^{-1}$) discussed in Section 2. 3, the uncertainty of the $O_2/N_2$ span sensitivity (±3%), and the uncertainty in the global averaged APO associated with the limited atmospheric sampling (±0.2 PgC yr$^{-1}$) discussed in Nevison et al. (2008) were also included in the calculation of the uncertainties in ΔAPO. The uncertainties of $\alpha_B$ and $\alpha_F$ were ±0.10 (Keeling and Manning, 2014) and ±0.04 (Tohjma et al., 2008), respectively. Finally, these uncertainties were propagated to the ocean and

land sink uncertainties in accordance with Eqs. (6) and (7).

We compared our global carbon budget estimations with those of GCP (Global Carbon Budget 2018) updated by Le Quéré, et al. (2018). In the GCP carbon budget assessment, the ocean and land sinks were estimated by combining multiple results from a variety of models including global ocean biogeochemistry models (GOBMs) and dynamic global vegetation models (DGVMs). Since the sum of the model-based ocean and land sinks was not necessarily balanced with the difference between

fossil fuel emissions and atmospheric accumulation, Le Quéré, et al. (2018) listed the discrepancies as budget imbalances. The ocean sinks, the land sinks which are 'net' land sinks computed as the differences between land uptake and emissions associated with land-use change, and the budget imbalances for the corresponding periods are listed in Table 2. Note that the uncertainties of the sinks of GCP are ±0.5 PgC yr$^{-1}$ for ocean and ±0.9PgC yr$^{-1}$ for land. The carbon sinks of this study and GCP for the three long periods are consistent with each other; the largest difference in sink strength is 0.35 PgC yr$^{-1}$, which is smaller than

the uncertainties associated with the individual estimations. A 3% higher span sensitivity of the $O_2/N_2$ measurements, which corresponds to the difference in the span sensitivity between SIO and NIES (section 3.1), would result in an increase and decrease of 0.27 PgC yr$^{-1}$ in the ocean and land sinks, respectively, for the three long periods. Although these changes would enlarge the differences in sinks between GCP and this study, the differences are still not significant given the uncertainties of both this study and GCP.

The carbon budgets for four pentad periods (2000-2004, 2004-2008, 2008-2012, 2012-2016) are also listed in Table 2. Here, we consider that the 5-year interval effectively reduced the apparent errors caused by the imbalance of the seasonal ocean $O_2$ fluxes, as is discussed in Section 3.1. Again, the discrepancies of the pentad ocean and land sinks between this study and GCP are within ±0.5 PgC yr$^{-1}$, which is also less than the estimated uncertainties. The land sink during 2008-2012 and the ocean sink during 2012-2016 of this study are about 0.5 PgC yr$^{-1}$ larger than those of GCP. These discrepancies in the carbon sinks,

partly explained by the rather large values of the carbon budget imbalances of the GCP estimation, might give a clue about how to partition the imbalance values between the land and ocean sinks.

Examining the temporal variations in the pentad sink strengths of this study, we found a gradual increase in the ocean sinks for the latter three pentad periods and a rapid increase and decrease in the land sinks for the former and latter two pentad

periods, respectively. The pentad averages of the GCP sinks seem to show similar temporal variations: a steady increase in the ocean sinks for the whole period and a rapid increase and decrease in the land sinks for the former and latter two pentad periods, respectively. These results suggest that the carbon sinks for the pentad periods can be used to evaluate the temporal changes. In the following section, we will examine the temporal change in the carbon sinks in more detail.

## 3. 3. Temporal change in the carbon sinks

Fig. 8 shows the temporal variations in the ocean and land sinks for the annual (broken red lines) and pentad (red lines) intervals calculated from the average of the APO changing rates based on the observations from HAT, COI, and cargo ships in the western Pacific. The uncertainties for the pentad sinks ($\pm 1\sigma$), which were calculated as described in the previous section, are shown as gray shading. To clearly understand the effect of $Z_{eff}$ correction, the pentad sinks without corrections are also depicted as purple lines in the figure. Additionally, the annual and pentad sinks of GCP are also depicted in the figures for comparison. Although the annual sinks show considerable variability especially for the first several years with peak-to-peak differences of more than 10 PgC yr$^{-1}$, the variability of the pentad sinks is effectively suppressed. Only the pentad budgets for 2000 show a rather large ocean uptake (3.21 PgC yr$^{-1}$) and a rather weak land emission (0.56 PgC yr$^{-1}$), which are depicted as dotted lines. These anomalous values may be explained by the fact that the influences from the considerable drawdown of APO in 2000-2001 cannot be compensated for in the pentad APO changing rate for 2000. Hamme and Keeling (2008) reported that the APO drawdown in 2000-2001, which was also observed in the SIO observations, may be attributed to deep ventilation associated with the unprecedented cooling of the western Pacific, and the variations in the ocean heat content exerted only secondary influence. Therefore, we don't use the anomalous pentad ocean and land sinks for 2000 in the following discussions.

The pentad ocean sinks show an overall increasing trend although there is a dip in the ocean sink centered on 2004-2005 by about 0.6 PgC yr$^{-1}$. Nevison et al. (2008) suggested that a decadal or longer period is needed to suppress the influence of the interannual variation in the ocean O$_2$ flux on the carbon sink estimation within $\pm 0.1$ PgC yr$^{-1}$ based on an ocean ecosystem model and an atmospheric transport model. In addition, the pentad APO changing rate still contains an uncertainty corresponding to $\pm 0.5$ PgC yr$^{-1}$ as is discussed in Section 3.1. Therefore, the anomalous dip in the ocean sink for 2004-2005 might be an error caused by the anomalous ocean O$_2$ flux variations. The increasing rate of the ocean sink during 2001-2014, determined by linear regression, is $0.08 \pm 0.02$ PgC yr$^{-2}$, which is larger than that of GCP which was $0.04 \pm 0.01$ PgC yr$^{-2}$. Although the temporal variability in the ocean sink in the GCP study is rather suppressed, which is attributed to the rather coarse resolution of the GOBMs (Le Quéré, et al., 2018), a much larger decadal and sub-decadal variability has been reported in the ocean sink estimations based on archived data of the observed surface partial pressure of CO$_2$ (pCO$_2$) (Landschützer et al., 2015; DeVries et al., 2017). Results from the Surface Ocean pCO$_2$ Mapping (SOCOM) initiative show that the decadal linear trend of the global ocean sink enhancement over 2001-2011 based on pCO$_2$ data and selected mapping methods is about 0.8 PgC yr$^{-1}$ per decade (Rödenbeck et al., 2015). For a detailed comparison, the global ocean sinks based on pCO$_2$ observations and interpolation techniques (Landschützer et al., 2016; Rödenbeck et al., 2014) for the period of 1990-2017 are

plotted in Fig. 9 together with the ocean sinks of both this study and GCP. Note that the extended $pCO_2$-derived ocean sinks were given as supplementary data of Le Quéré, et al. (2018) and those sinks were uniformly inflated by 0.78 PgC yr$^{-1}$ to compensate for the pre-industrial steady state source of $CO_2$ derived from riverine input of carbon to the ocean (Resplandy et al., 2018). Both the GCP and $pCO_2$-derived ocean sinks show changes in the trends before and after 2001, while the magnitude of the changes in the $pCO_2$-derived sinks are larger. The increasing rates determined by linear regression during 2001-2014 are $0.08 \pm 0.01$ PgC yr$^{-2}$ for Landschützer et al. (2016) and $0.07 \pm 0.02$ PgC yr$^{-2}$ for Rödenbeck et al. (2014), which are more consistent with the rate found in this study. Therefore, our result seems to support a previous conclusion that the recent increase in the ocean sinks exceeds the increasing trend of ocean sink expected only from the atmospheric $CO_2$ increase (Landschützer et al., 2015; DeVries et al., 2017).

In contrast, the pentad land sinks of both this study and the GCP study show an increasing trend during 2001-2009 followed by a decreasing trend during 2009-2014, although the range of variations of this study is about two times larger than that of GCP. The linear trends for the former period are $0.23 \pm 0.04$ PgC yr$^{-2}$ for this study and $0.10 \pm 0.03$ PgC yr$^{-2}$ for GCP, and those for the latter period are $-0.22 \pm 0.04$ PgC yr$^{-2}$ for this study and $-0.12 \pm 0.04$ PgC yr$^{-2}$ for GCP. An enhancement of the land uptake during the 2000s has been reported recently by several studies based on atmospheric inversions and biosphere models (Keenan, et al., 2016; Kondo et al., 2018; Piao et al., 2018). Although there is an ongoing discussion about the detailed mechanisms of the enhanced net land uptake, the accelerated land uptake may partially explain the stagnation of the growth rate of the atmospheric $CO_2$ in the 2000s despite the increasing anthropogenic $CO_2$ emissions. Examining the atmospheric inversion studies and the previous version of GCP (Le Quéré, et al., 2015), in which the net land uptake were computed as residuals among the other carbon budget components, Piao et al. (2018) found that the linear increasing trend of the net land carbon sink during 1998-2012 was $0.17 \pm 0.05$ PgC yr$^{-2}$. The linear trend of this study during 2001-2009 is close to the above value within the uncertainty. Although the corresponding linear trend of the latest GCP estimation is about half of that of the present study, the sum of the net land sink and the budget imbalances of GCP, plotted as light blue lines in Fig. 8, shows a much larger increasing trend, $0.20 \pm 0.03$ PgC yr$^{-2}$, which is almost identical to our trend.

The land sinks of both this study and the GCP study exhibit decreasing trends for the period 2009-2014, which are partially compensated for by the steady increase in the ocean uptake. The atmospheric accumulation rate of $CO_2$ significantly increased in 2015 and 2016 (see Figure 5), when one of the strongest El Niño events occurred. Studies based on atmospheric $CO_2$ observations from stations and by satellite indicated that a reduction in biospheric uptake and an increase in biomass burning contributed to the $CO_2$ increase during the El Niño event (Chatterjee, et al., 2017; Patra et al., 2017). The decreasing trend of the pentad land uptake also reflects the change in the global carbon cycle associated with the El Niño event.

Shown as discrepancies between the pentad sinks of GCP with and without budget imbalances (Fig. 8), the magnitude of the budget imbalances seems to increase after 2007. For the pentad sinks centered on 2007, 2008 and 2009, the ocean and land sinks of this study agree with those of GCP without and with the budget imbalances, respectively. For the pentad sinks between 2010 and 2014, both the ocean and land sinks of this study are plotted between those of GCP with and without the budget

imbalances. Although we cannot show any definitive partitioning of the budget imbalance between ocean and land sinks because of a rather large uncertainty associated with the sink estimations, the above results seem to suggest that the budget imbalances are allocated to land sinks for the former period and to both sinks for the latter period.

From the above discussions, we feel that a five-year duration effectively suppresses to some extent the anomalous variations in the carbon budget estimations based on APO, which are considered to be caused by the imbalance of the seasonal air-sea $O_2$ exchange. Probably, the five-year average suppresses the variability of $Z_{eff}$ to a level of $\pm 0.5$ PgC yr$^{-1}$, as is discussed in Section 3.1. To reduce uncertainty in the carbon budget estimation, we need more effort to improve the quantification of the net $O_2$ outgassing associated with global ocean warming because the quantification of the $Z_{eff}$ at this state is still very speculative. Applying the approach of Stendardo and Gruber (2012), who examined the long-term changes in dissolved $O_2$ and heat content by using archived oceanographic data of the Atlantic Ocean, to other ocean basins would improve our understanding of the long-term net ocean $O_2$ flux/heat flux ratio.

## 4. Conclusion

We have evaluated the global carbon budgets based on the APO data computed from the $O_2/N_2$ and $CO_2$ of the flask samples collected in the Pacific region since 1997. In the carbon budget calculation, we corrected the ocean and land sinks with the ocean $O_2$ outgassing effect, $Z_{eff}$, based on the ocean heat increment for the 0-2000 m layer. Eventually, we have obtained the following conclusions:

1) The long-term oceanic and land biotic carbon sinks were $2.6 \pm 0.7$ PgC yr$^{-1}$ and $1.5 \pm 0.9$ PgC yr$^{-1}$, respectively, for a 17-year period (2000-2016), and $2.4 \pm 0.7$ PgC yr$^{-1}$ and $1.9 \pm 0.9$ PgC yr$^{-1}$, respectively, for a 14-year period (2003-2016). These long-term carbon sinks agreed well with those of the latest GCP estimation (Le Quéré, et al., 2018); the differences of the individual estimations are less than $\pm 0.35$ PgC yr$^{-1}$.

2) The ocean and land sinks for the four pentad (five-year) periods (2000-2004, 2004-2008, 2008-2012, 2012-2016) of this study also showed good agreement with those of GCP within a difference of $\pm 0.5$ PgC yr$^{-1}$. The land and ocean sinks of this study showed larger values by about 0.5 PgC yr$^{-1}$ than those of GCP for 2008-2012 and 2012-2016, respectively, when rather large carbon budget imbalances (>0.5 PgC yr$^{-1}$) were found. Therefore, the discrepancies in the sinks between this study and GCP might give a clue about how to partition the imbalance values between the land and ocean sinks.

3) Calculating the carbon budgets for the pentad periods consecutively, we examined the changing trend of the ocean and land sinks during a 14-year period (2001-2014). In general, the changing trends of both land and ocean sinks of this study agreed well with those of GCP, although the range of variations of this study was about two times larger than that of the GCP study. The pentad ocean sinks showed an overall increasing trend for the entire period (2001-2014) with a linear increasing rate of $0.08 \pm 0.02$ PgC yr$^{-2}$. This increasing rate was about two times larger than that

for the GCP ocean sinks ($0.04 \pm 0.01$ PgC yr$^{-2}$) but was consistent with those for the global ocean sinks based on $pCO_2$ observations and interpolation techniques (Landschützer et al., 2016; Rödenbeck et al., 2014). In contrast, the pentad land sinks showed an increasing trend for 2001-2009 and a decreasing trend for 2009-2014. The linear trends of the land sinks for this study and the GCP (in parentheses) were $0.23 \pm 0.04$ PgC yr$^{-2}$ ($0.10 \pm 0.03$ PgC yr$^{-2}$) for the former period and $-0.22 \pm 0.04$ PgC yr$^{-2}$ ($-0.12 \pm 0.04$ PgC yr$^{-2}$) for the latter period. Enhancement of the land carbon uptake was reported also by previous studies (Keenan, et al., 2016; Kondo et al., 2018; Piao et al., 2018). In addition, the recent decreasing trend of the land uptake was found to be partially related to the global carbon cycle variation associated with the strong El Nino event in 2015 and 2016.

## 5. Code/Data availability

All the observed data needed to evaluate the global carbon budget are provided in the tables in the supplement. Other data are available upon request to the corresponding author (tohjima@nies.go.jp).

## 6. Author contributions

YT designed the study and drafted the manuscript. YT and YH conducted the $O_2/N_2$ measurements and analyzed the observed data. HM and TM organized the flask sampling at the monitoring stations and conducted the $CO_2$ measurements. SN managed the shipboard flask sampling. HM, TM, SN, and YH examined the results and provided feedback on the manuscript. All the authors approved the final manuscript.

## 7. Competing interests

The authors declare that they have no conflict of interest.

## 8. Acknowledgments

We thank Nobukatsu Oda, Fujio Shimano, Kousei Yumoto, Shigeru Kariya, Tomoyasu Yamada and other members of the Global Environment Forum (GEF) for their continued support in flask sampling of air samples. We would like to acknowledge the assistance of the caretakers and local staff of the Ochiishi and Hateruma monitoring stations as well as the owners, operators and crew of the cargo ships. We are grateful to Hisayo Sandanbata, Eri Matsuura, Yoko Kajita, and Motoki Sasakawa of the National Institute for Environmental Studies for their continued support in the $CO_2$ measurements of the flask samples. We also thank Keiichi Katsumata for preparing the working standard gases for the $O_2/N_2$ measurements. We gratefully acknowledge the generous cooperation of Toyofuji Shipping Co. and Kagoshima Senpaku Co. for providing us the opportunity to make the aboard atmospheric observations. We would like to thank the two anonymous referees for helping us to greatly

improve the manuscript.

## 9. Financial support.

This research was financially supported by the Global Environmental Research Coordinate System from the Ministry of the Environment, Japan (E0955, E1451)

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

**Tables**

**Table 1.** Comparison of changing rate of the atmospheric $CO_2$, $O_2$, and APO

| Period | Site | Average changing rate | | | Ref. |
|---|---|---|---|---|---|
| | | $CO_2$ (ppm yr$^{-1}$) | $O_2$ (per meg yr$^{-1}$) | APO (per meg yr$^{-1}$) | |
| 1998-2016 | HAT | 2.19 ± 0.01 | −21.8 ± 0.8 | −10.4 ± 0.8 | This study |
| 2000-2016 | HAT | 2.21 ± 0.02 | −22.0 ± 0.8 | −10.4 ± 0.8 | This study |
| 2000-2016 | COI | 2.22 ± 0.02 | −21.9 ± 0.8 | −10.2 ± 0.8 | This study |
| 2003-2016 | HAT | 2.25 ± 0.02 | −21.8 ± 0.8 | −10.1 ± 0.8 | This study |
| 2003-2016 | COI | 2.26 ± 0.02 | −21.6 ± 0.8 | −9.8 ± 0.8 | This study |
| 2003-2016 | W-Pacific | 2.15 ± 0.06 | −21.2 ± 0.8 | −10.0 ± 0.8 | This study |
| 2000-2009 | HAT | 2.04 ± 0.02 | −20.4 ± 0.8 | −9.7 ± 0.8 | This study |
| 2000-2009 | COI | 1.91 ± 0.03 | −19.9 ± 0.8 | −9.7 ± 0.8 | This study |
| 2000-2009 | Global | 1.90 ± 0.02 | - | −10.4 ± 0.5 | Keeling & Manning (2014) |
| 2001-2009 | HAT | 2.08 ± 0.03 | −20.0 ± 0.8 | −8.9 ± 0.8 | This study |
| 2001-2009 | COI | 1.87 ± 0.03 | −19.0 ± 0.8 | −9.2 ± 0.7 | This study |
| 2001-2009 | Ny-Ålesund | 2.00 ± 0.08 | −21.2 ± 0.8 | - | Ishidoya et al. (2012) |
| 2001-2009 | Showa | 1.99 ± 0.06 | −22.0 ± 0.8 | - | Ishidoya et al. (2012) |
| 2001-2013 | HAT | 2.19 ± 0.02 | −21.3 ± 0.8 | −9.7 ± 0.8 | This study |
| 2001-2013 | COI | 2.07 ± 0.03 | −20.4 ± 0.3 | −9.6 ± 0.8 | This study |
| 2001-2013 | Ny-Ålesund | 1.99 ± 0.02 | −19.9 ± 0.3 | −10.1 ± 0.3 | Goto et al. (2017) |
| 2001-2013 | ALT, MLO, SPO | 1.98 ± 0.03[a] | −20.5 ± 0.3[a] | −10.8 ± 0.1[a] | Goto et al. (2017) |

[a]Average and standard deviation of the changing rates for the three sites (ALT, MLO and SPO) listed in Table 1 of Goto et al. (2017) are given in this table.

**Table 2.** Comparison of global carbon budgets based on APO with those from GCP[a,b]

| Period | $\Delta$APO[c] | Atm. $CO_2$[d] | Fossil fuel[d] | $\alpha_F$[e] | $Z_{eff}/1.1$[f] | Sink of this study Ocean | Sink of this study Land | Sink of GCP[d] Ocean | Sink of GCP[d] Land | Imb. |
|---|---|---|---|---|---|---|---|---|---|---|
| 1998-2016 | -10.3(0.91) | 4.45 | 8.28 | 1.38 | 0.52 | 2.57(0.71) | 1.26(0.89) | 2.24 | 1.46 | 0.13 |
| 2000-2016 | -10.3(0.91) | 4.45 | 8.48 | 1.38 | 0.54 | 2.55(0.73) | 1.48(0.91) | 2.27 | 1.48 | 0.29 |
| 2003-2016 | -9.9(0.91) | 4.58 | 8.83 | 1.38 | 0.52 | 2.35(0.73) | 1.90(0.93) | 2.34 | 1.55 | 0.36 |
| 2000-2004 | -8.8(0.94) | 3.93 | 7.11 | 1.40 | 0.59 | 2.23(0.76) | 0.94(0.90) | 2.01 | 1.30 | −0.14 |
| 2004-2008 | -9.2(0.96) | 4.08 | 8.21 | 1.38 | 0.33 | 1.97(0.62) | 2.17(0.82) | 2.18 | 1.74 | 0.22 |
| 2008-2012 | -10.4(0.98) | 4.19 | 9.05 | 1.37 | 0.54 | 2.54(0.77) | 2.31(0.97) | 2.32 | 1.85 | 0.68 |
| 2012-2016 | -11.6(1.06) | 5.36 | 9.65 | 1.37 | 0.71 | 3.05(0.90) | 1.25(1.09) | 2.55 | 1.26 | 0.49 |

[a]Figures are given in units of per meg $yr^{-1}$ for $\Delta$APO, mol $mol^{-1}$ for $\alpha_F$, and PgC $yr^{-1}$ for the others.

[b]Figures in parentheses represent the uncertainties.

[c]$\Delta$APO is based on the data from HAT, COI, and cargo ships (40°S-30°N). The uncertainty in parentheses includes the uncertainty associated with the observations, stability in the $O_2/N_2$ scale, uncertainty derived from limited sampling, and uncertainty in the $O_2/N_2$ span sensitivity (see text).

[d]These figures are computed from the dataset summarized by the Global Carbon Project (GCP). The uncertainties are ±0.2 PgC $yr^{-1}$ for the atmospheric $CO_2$ and ±5% for the fossil fuel emissions, ±0.5 PgC $yr^{-1}$ for the ocean sinks, and ±0.9PgC $yr^{-1}$ for the land sinks (Le Quéré, et al., 2018).

[e]The uncertainties for $\alpha_F$ are ±0.04 mol $mol^{-1}$ (Tohjima et al., 2008).

[f]The values of $Z_{eff}$ include both global ocean warming and anthropogenic nitrogen deposition effects, and the uncertainties are assumed to be ±100% (see text).

**Figure captions**

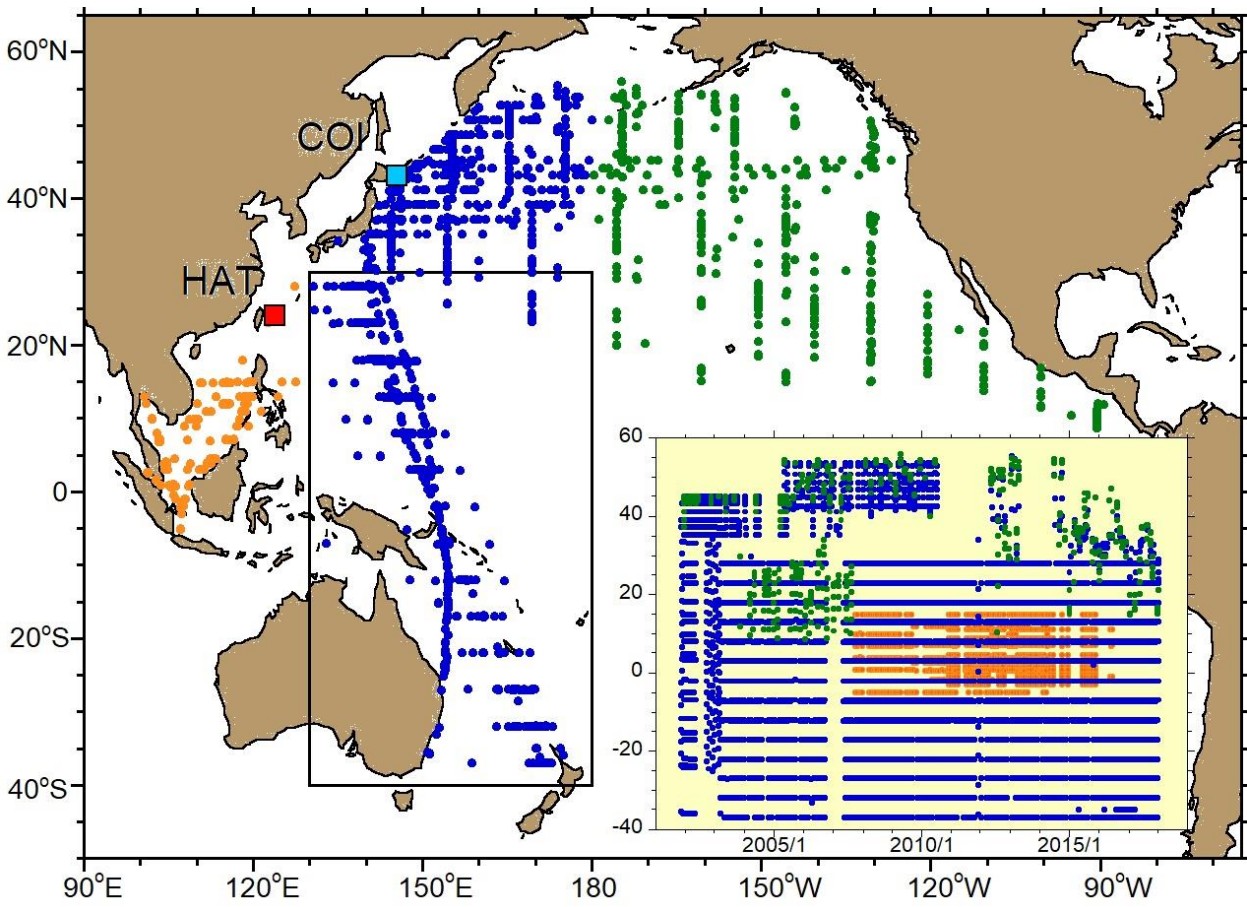

**Fig. 1.** Map showing the air sampling locations in the Pacific region. The light blue and red squares represent the monitoring stations of COI and HAT, respectively. The orange, blue, and green circles correspond to the positions where flask samplings were taken onboard cargo ships in South East Asia, western Pacific, and eastern Pacific, respectively. The inserted figure shows the time latitude distribution of onboard flask samples. The flask data from COI, HAT and the regions in the black rectangle (130-180°E, 40°S-30°N) were used in the budget calculations.

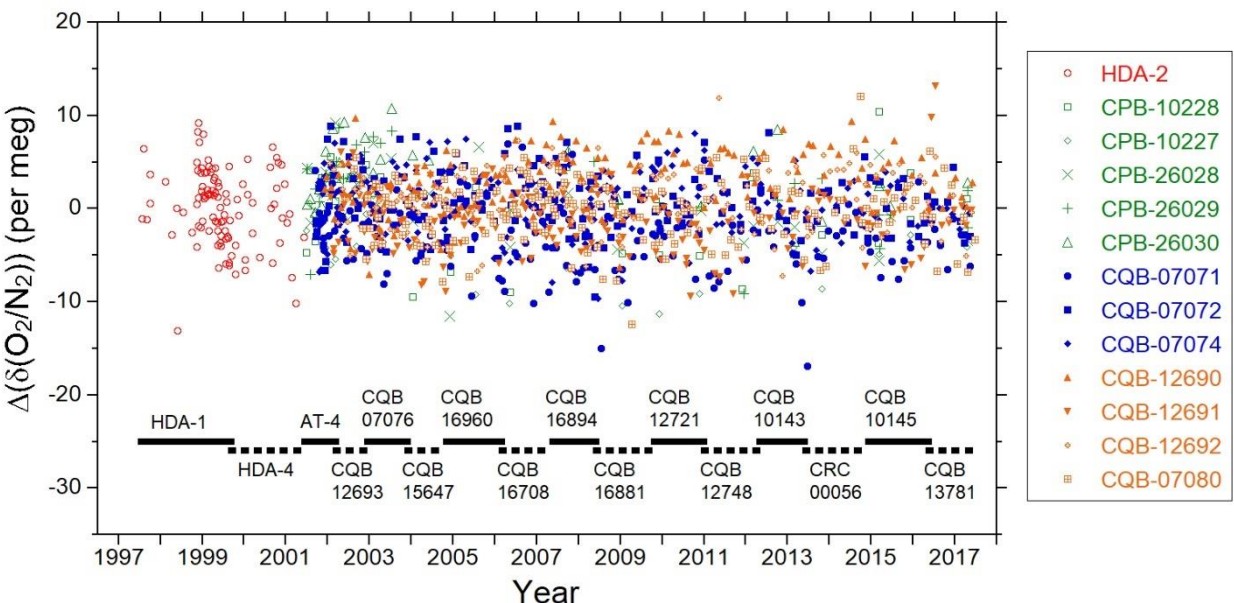

**Fig. 2.** Temporal changes in the $O_2/N_2$ ratio of primary reference gases relative to the NIES $O_2/N_2$ scale. The differences of the $O_2/N_2$ ratio from the average are plotted for HDA-2 along with the differences of the $O_2/N_2$ ratio from the initial values for the individual cylinders except HDA-2. Solid and broken horizontal bars in the lower part of the figure indicate the periods when the working reference gases were used.

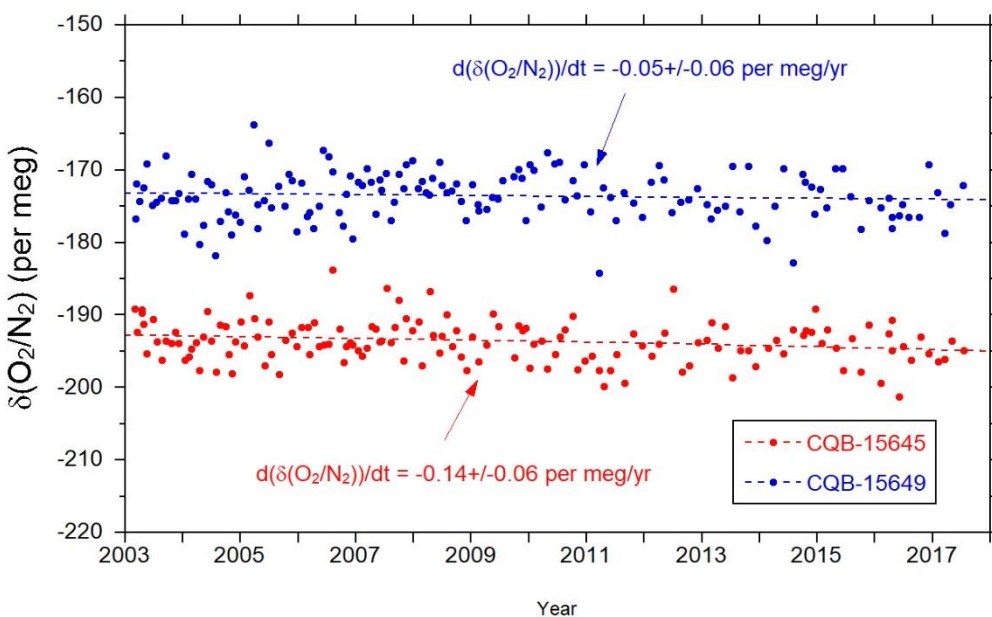

**Fig. 3.** Temporal changes in the $O_2/N_2$ ratio of reference airs in two aluminum cylinders which are independent from the NIES primary reference gases, relative to the NIES $O_2/N_2$ scale. The broken lines represent the linear regression lines.

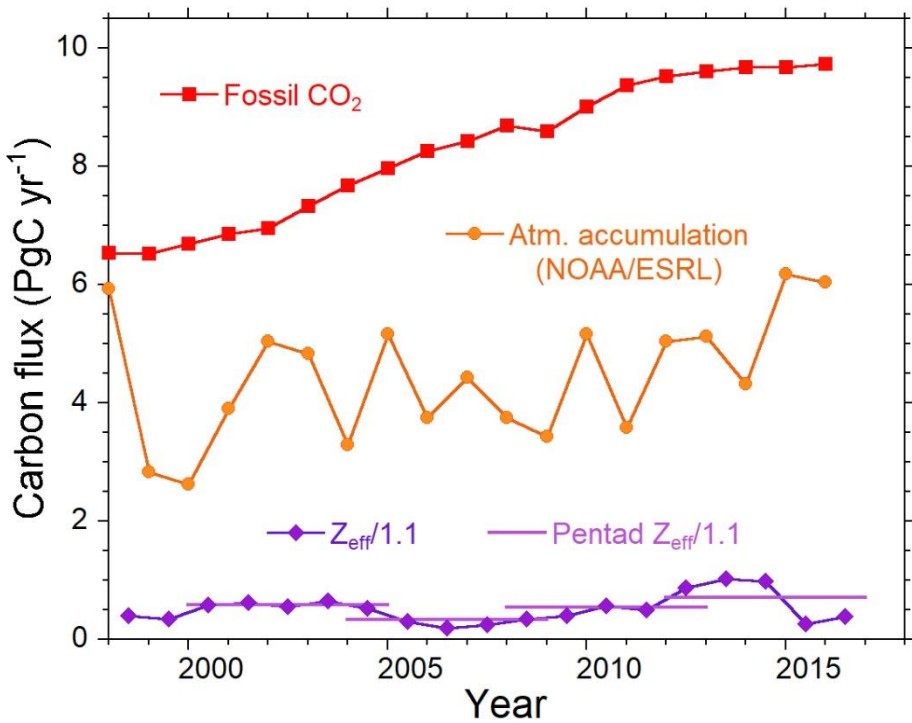

**Fig. 4.** Temporal changes in fossil fuel $CO_2$ emissions (red squares), atmospheric $CO_2$ accumulation rate (orange circles), and ocean outgassing effect $Z_{eff}$ divided by land biotic $-O_2/CO_2$ exchange rate (1.1) (purple diamonds). The 5-year averages of $Z_{eff}/1.1$ used for the pentad (five-year) carbon sink calculations are also depicted as purple lines.

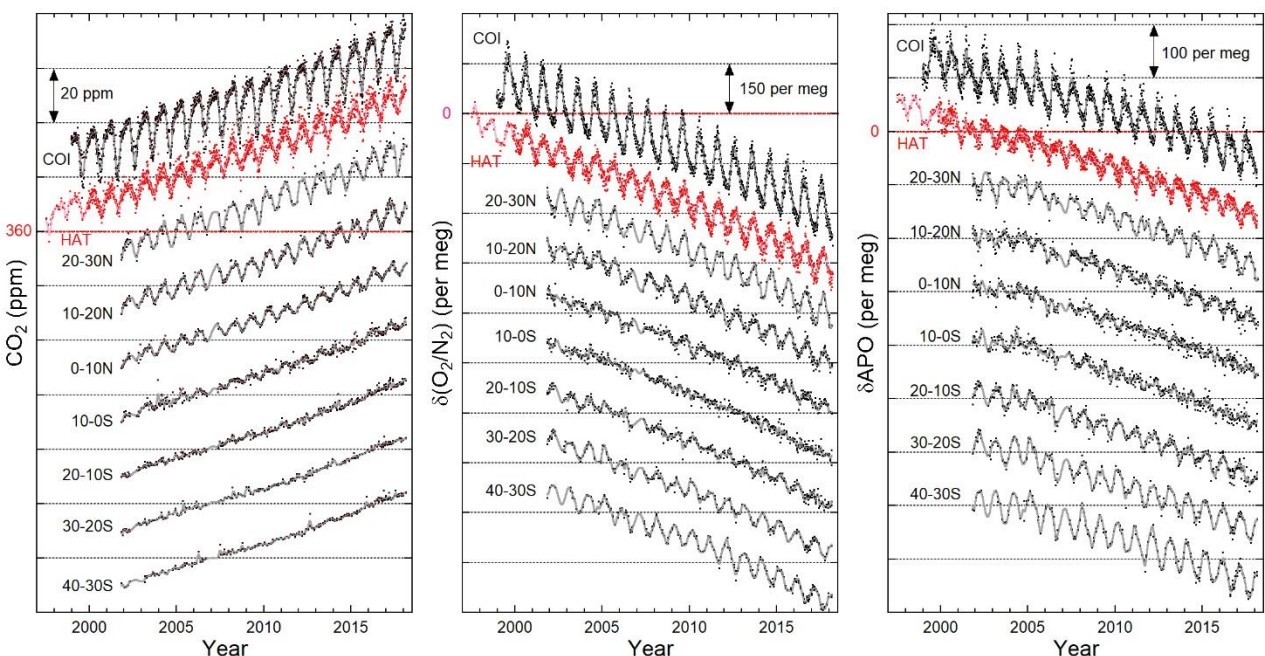

**Fig. 5.** Time series of the atmospheric $CO_2$ mole fraction (left), $O_2/N_2$ ratio (middle), and APO (right) of the flask samples obtained from the NIES flask sampling network shown in **Fig. 1**. Observed data from COI, HAT, and cargo ships operating between 40ºS and 30ºN were used for the global carbon budget calculation. The time series of $CO_2$, $O_2/N_2$ and APO are offset by 20 ppm, 150 per meg, and 100 per meg, respectively, to allow them to be plotted on the same panels. The numbers on the y-axis represent the values for the data at HAT.

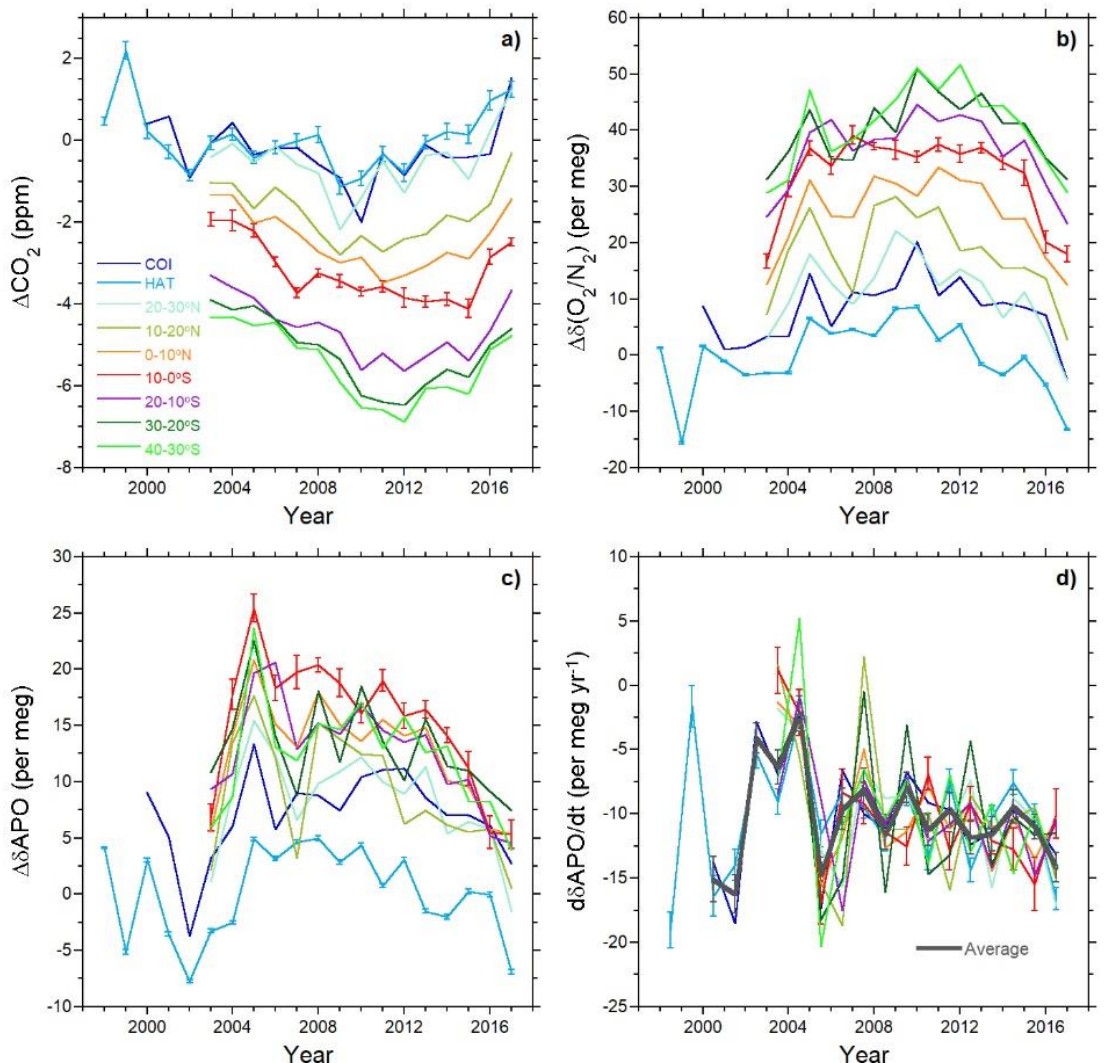

**Fig. 6.** Temporal variations of the (a) annual mean $CO_2$, (b) annual mean $O_2/N_2$, (c) annual mean APO, and (d) annual changing rate of APO based on the flask samples collected from HAT, COI, and cargo ships in the western Pacific (40°S-30°N). The differences in the annual means from the linear trends fitted to the data at HAT are depicted in the figures to emphasize the interannual variations. Vertical bars for the plots of HAT and 10-0°S bin correspond to the standard errors of the annual means.

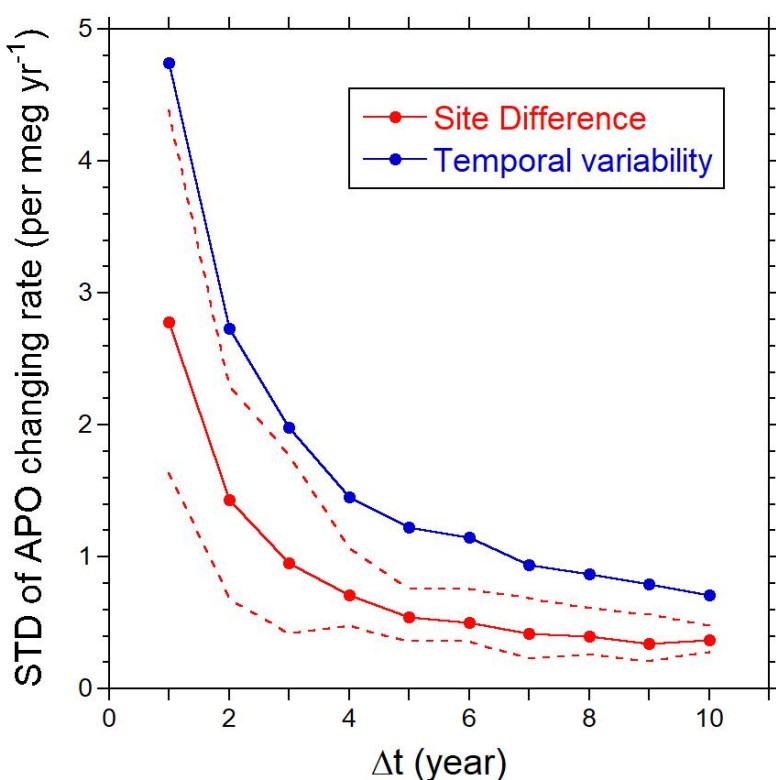

**Fig. 7.** Relationship between the standard deviation of the APO changing rate and the time interval to calculate the changing rate. The red circles represent the averages of the standard deviations of the APO changing rates from the different sites for the same year. The broken lines represent the minimum and maximum of the standard deviations. The blue circles represent the temporal variability of the average APO changing rate of the different sites.

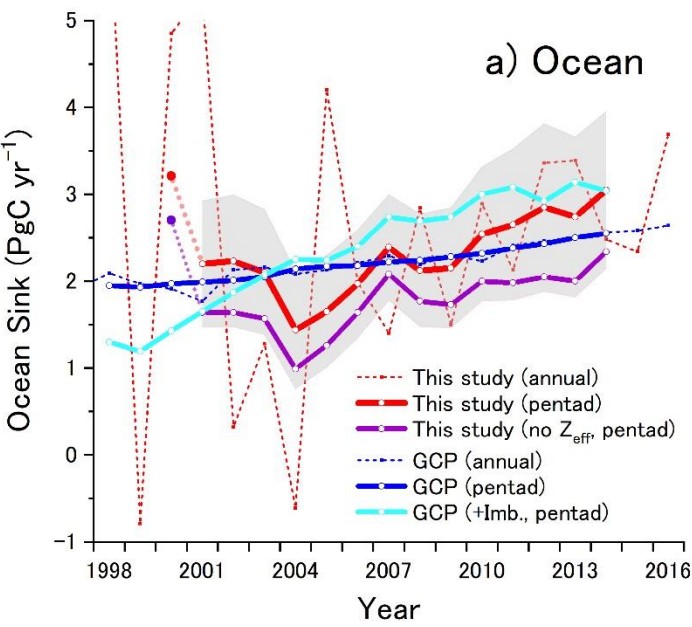

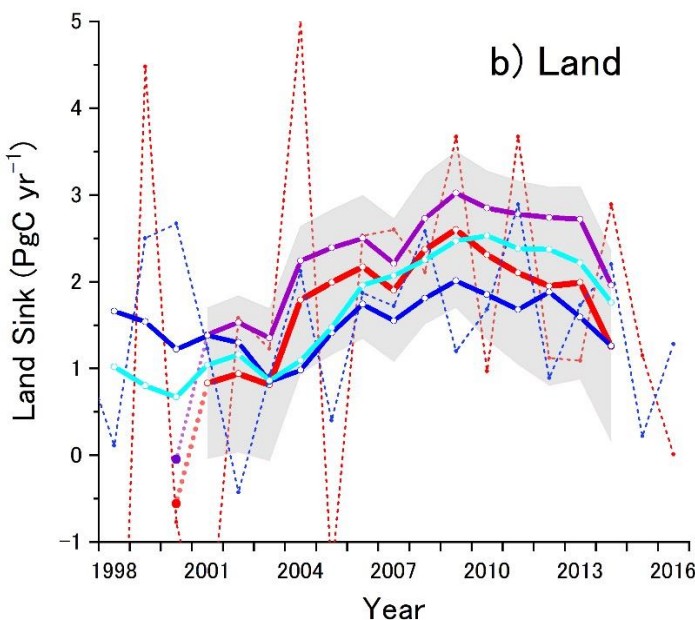

**Fig. 8.** Temporal variations in (a) ocean and (b) land biospheric sinks estimated from APO variations of this study (red) and process-based models of GCP (blue). The thin broken lines represent the annual sinks and the thick lines represent the pentad

sinks. The purple lines represent the pentad sinks based on APO without ocean outgassing correction ($Z_{eff}$) and the light blue lines represent the sinks of GCP with the imbalance sinks added. The uncertainty associated with the pentad sinks with $Z_{eff}$ corrections are shown as shaded area.

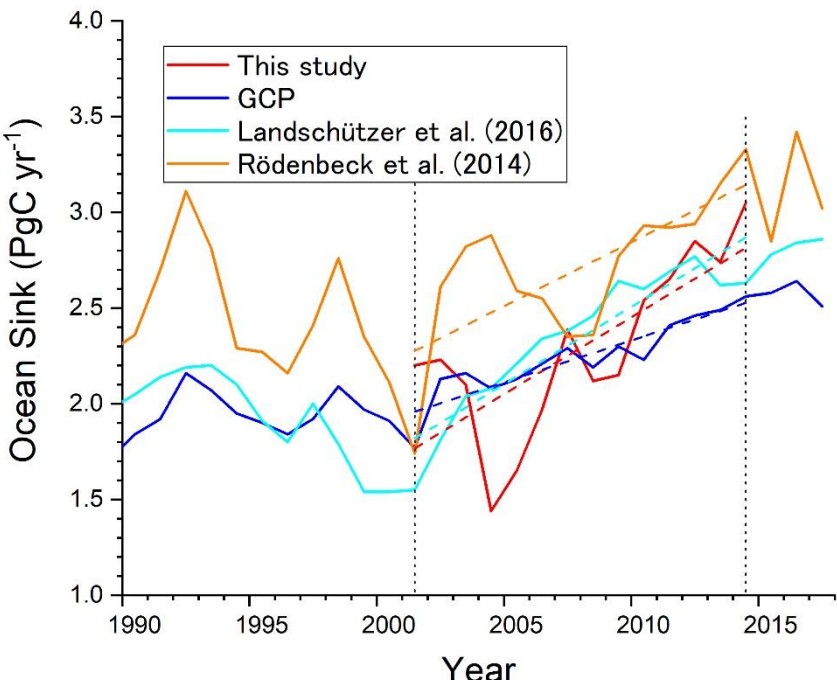

**Fig. 9.** Comparison of the temporal variations of the ocean sinks based on the APO data of this study (red), global ocean biogeochemistry models (GOBMs) of GCP (blue), and $pCO_2$ data of Landschützer et al. (2016) (light blue) and Rödenbeck et al. (2014) (orange). The broken lines represent the regression lines for the corresponding data during 2001-2014. Note that the $pCO_2$-based ocean sinks are adjusted for the pre-industrial ocean $CO_2$ emissions ($\pm 0.78$PgC yr$^{-1}$) caused by riverine $CO_2$ input to the ocean (Resplandy et al., 2018).