# Peer review of "Global carbon budgets estimated from atmospheric $O_2/N_2$ and $CO_2$ observations in the western Pacific region over a 15-year period"

_Atmospheric Chemistry and Physics, 2019_

## Referee Comment (RC1) · Anonymous Referee #1 · 17 Apr 2019

In this paper, the authors present recent global CO2 budgets based on precise measurements of atmospheric O2/N2 and CO2 in the western Pacific region. The data quality is excellent and the authors' effort to maintain the long-term observation over the wide area is highly commended. Although the paper does not provide particularly surprising new findings such as the equatorial bulge and mid-latitudinal trough of APO reported by the authors' past studies, the data itself is noteworthy and the presented results constitute an important contribution towards an independent validation of the global carbon cycle reported by Global Carbon Project (GCP). The paper is well written and clear. I recommend this paper for publication in ACP with a few additional minor revisions below.

[Figure]

1) The uncertainties of F and Z_eff should be presented explicitly somewhere in the main text and/or Tables. Did the authors assume the uncertainty of 100% for Z_eff following Keeling and Manning (2014)? Also, for those readers not familiar with O2/N2 studies, it would be better to present the representative values of alpha_F for the period 1998-2016 or the respective values for the periods in Table 2.

2) The discussion on the evaluation of the needed interval to suppress the temporal variability in Z_eff is useful in deriving reasonable interannual variations in CO2 sinks from O2/N2 observations. The needed interval was estimated to be 5 years in the paper, and the authors used the ocean heat storage of 0-2000 m layer to estimate Z_eff based on the gas flux / heat flux ratio reported by Keeling and Garcia (2002). However, I think the circulation time of ocean deep layer water is much longer than 5 years. Please explain why the authors have considered the use of heat storage of 0-2000 m to be more reasonable than that of 0-700 m. I suppose there is an implicit assumption in the analysis that the ocean circulation and oxygen concentration are in steady-state from the surface to the deep layer. However, temporal variations found in the 5-years average Zeff in Table2 suggest that the ocean is not in steady-state.

3) I think it may be helpful for the reader to note the differences in land and ocean CO2 uptakes expected from the 3% difference in the span sensitivity between NIES and SIO. Has the conclusion about the comparison of the CO2 uptake reported by the present study with those by GCP changed significantly due to the difference in the span sensitivity?

4) The authors have compared land and ocean CO2 sinks estimated in the present study with those obtained by GCP, with and without the imbalance sinks added. It seems to me that the authors conclude that the differences between the present study and GCP are reduced, both for land and ocean CO2 sinks, by adding the "total" imbalance to the respective sinks. However, I actually think we can only add the imbalance to the land and ocean CO2 sinks based on an appropriate differential distribution. I understand it would be difficult to suggest the best distribution due to uncertainties of

the estimated CO2 sinks, but I would like to hear the authors' thoughts on this.

5) P3, line 32: A literal error "...heat content,." should be corrected.

6) P7, line 19: I think the unit of Z_eff is not TgC yr-1 but PgCyr-1 in this context.

7) References: Please consolidate the format of references. For example, some journal titles are written in Italic and the others are not.

8) Caption in Fig.7: The phrase "changing ratio" should be changed to "changing rate".
* * *

---

## Referee Comment (RC2) · Anonymous Referee #2 · 13 May 2019

Summary: Tohjima et al. here present independent estimates of global carbon sinks using a long-term APO record based on O2/N2 and CO2 measurements over the 2000-2016 period from two stations in Japan and cruise ship measurements from the western Pacific. Using these long-term APO time series, fossil fuel records, global CO2 time series, and correcting for changes in O2 fluxes due to changes in ocean heat content (Z_eff), they calculate the ocean and land sinks of anthropogenic carbon on decadal and pentadal timescales, and find mean values and variability in trends that are relatively in good agreement (within uncertainties) with the GCP reported sinks estimates. Overall, this study is highly relevant and important to the carbon cycle community, providing independent estimates of carbon sinks and thus can help inform observational

estimates and test models used to infer past and future projections of the global carbon cycle. The paper is well written and structured, though it could benefit from one more round of proofreading for clarity in certain sections (e.g. introduction and conclusion). The paper shows interesting and highly relevant findings that are suitable for publication in the journal of ACP, though I have a few concerns regarding the treatment of uncertainty outlined below that I hope the authors can resolve/clarify. Further, these time-series have great potential to be used as an independent check on specific modeling and ocean observations-based products submitted to the GCP, rather than solely compared to the GCP mean which shows large variations across products. This latter point may be outside the current scope of this work, but could significantly improve the impact of this interesting and relevant study.

Main Issues: The main issues identified in this paper are listed as follow:

1) It's unclear how uncertainty in estimating $Z_{eff}$ comes into play in the carbon budget estimates, especially given its relevance for the shorter timescales considered. The authors do show trends without $Z_{eff}$ (figure 8) but it's not evident if this is incorporated in the carbon sinks and trends calculation (i.e. unclear if incorporated in grey shading in figure 8 or error estimates in carbon sinks column in Table 2). Perhaps, the authors could evaluate uncertainty in $Z_{eff}$ from using the upper/lower bounds with and without $Z_{eff}$? Further, it's unclear how correcting for $Z_{eff}$ in the carbon budgets plays out in the 5 year timescales described (as also discussed in Nevison et al. 2008 and elsewhere). It seems, as referred to by the authors, that the ventilation events of 1999-2001 could impact the pentad trends, and thus similar variability during other years probably could have similar effects on other pentad periods (e.g. 2004-2005 dip in pentad ocean sink seems to co-occur with inter-annual variation that don't seem to be fully suppressed?). Finally, an additional and not insignificant component of $Z_{eff}$ not included in this study is the atmospheric deposition effect, as detailed in Keeling and Manning 2014, which adds about 0.1 (+/-0.1) Pg C/yr, and which should raise $Z_{eff}$ from 0.1-0.9 Pg C/yr, to 0.2-1.0 Pg C/yr. Overall, I fell the treatment of $Z_{eff}$ uncertainty

within the shorter timescales considered here merits further clarification.

2) I see the need perhaps for a section dedicated to clarifying and detailing the sources, contribution, and methods for calculating carbon budget uncertainty, as it can help clarify the confidence in the pentad trends and conclusions presented here. Table 2, for instance, could incorporate uncertainty due to Z_eff in the carbon sinks column, and in Figure 8 (gray shading). Uncertainty due to undersampling the global signal has also been shown by Nevison et al. (2008) to contribute to uncertainty in estimating budgets on the shorter timescales evaluated here. What is impact of sampling over the western Pacific (∼40S-40N) vs. full global sampling on the carbon sink trends? How does uncertainty in alpha_B in using values of 1.1 vs 1.05 affect the uncertainty in carbon sink budgets? Finally, it's unclear how the contribution of measurement uncertainty, due to span calibration of the gas chromatographic technique and potential longterm drift shared across all cylinders, is incorporated in the uncertainty analysis.

3) The comparison against the GCP could be elaborated on a bit more, as it raises important issues in the field. The authors could elaborate further (through existing or new figure/table) how different estimates reported by GCP compare to the APO method, including hindcast ocean models and ocean observation based products, all of which are readily available in the GCP product as globally integrated fluxes: https://www.icos-cp.eu/GCP/2018. It is interesting that the comparison to the GCP mean showcases similarities in magnitude and in temporal evolution of pentads. The point that the uptake of carbon by the ocean is larger than expected from atmospheric increase alone is very interesting. How do the decadal trends (2000-2016) in this study compare to the pCO2 based air-sea flux timeseries by Landshutzer et al (2016) and Rodenbeck et al (2013), as both of these estimates seem to show larger decadal variability than the ocean models? These items may be beyond the current scope of this study, but could substantially improve the impact of this paper with (hopefully?) relatively minor figure/text additions.

Minor Issues: Minor issues, edits, typos, and technical issues are listed below:

Pg2 L27: "The estimated value for $\alpha$F is about 1.10$\pm$0.05 (Severinghaus, 1995) and that for $\alpha$B is about 1.4 (Keeling, 1988)." Should be the other way around: $\alpha$B is 1.10 and $\alpha$F is 1.4.

Perhaps add citations for Equations (1), (2), and (3)?

P3 L1, this paragraph could use a brief explanation of APO concept as a tracer for those not familiar with APO, i.e. cancellation of terrestrial influence, etc.

Pg 7 L29, shouldn't Z_eff be in PgC/yr?

Pg 9 Line 20, the ENSO topic deserves a bit more clarification here. It would be good to preface the ENSO sentence with the findings of Rodenbeck et al 2008, who suggest anomalous outgassing of APO during El Niño, while Tohjima et al 2015 show a suppressed peak instead, and clarify that Eddebbar et al (2017) reconcile this apparent discrepancy through a model-simulated zonal dipole-like ENSO response in the equatorial Pacific, and that enhanced observational zonal coverage in this region is needed to constrain the full basin ENSO response.

Suggested editing notes:

Pg 2 Line 5: remove "still", and add year by which emissions rose to 10 Pg C/yr?

Pg 2 Line 6: "Paris Agreement ... aimed to balance the anthropogenic greenhouse gas emissions and natural removals in the second half of this century...", I suggest editing to: " ... aimed to reduce anthropogenic greenhouse gas emissions to maintain the increase in global mean surface temperature well below 2$^\circ$C by 2100, ..."?

Pg3 L8, suggest deleting "In these days".

Pg3 L21, "which reduces the ventilation of the seawater.", suggest instead: "which reduces the ventilation of interior water masses."

Pg3 L26: replace "huge" with "large"

Pg 14 L 20: Not sure I understand this sentence: "This means that the changing trends of carbon budgets may be evaluated by the at least decadal APO data." suggest rephrasing and/or elaborating further?

Pg 12 L 32. Replace "stagnant" with "stagnancy"

Pg13 L 1: replace "in spite of' with "despite"

---

## Author Comment (AC1) · 21 Jun 2019

**Response to review comments on "Global carbon budgets estimated from atmospheric $O_2/N_2$ and $CO_2$ observations in the western Pacific region over a 15-year period"**

**Anonymous Referee #1:**
We would like to thank the anonymous referee #1 for his/her helpful comments and suggestions on our paper. We have revised the manuscript as is described in the following. The referee's comments are in *blue italics*, and the modifications are shown in red.

First, we would like to mention about the GCP-reported data used in this paper. In the original manuscript, we used fossil fuel emissions, atmospheric accumulation, and global sink estimates taken from Global Carbon Budget 2017 reported by GCP (Le Quéré, et al., 2018). However, the updated data of Global Carbon Budget 2018 (Le Quéré, et al., 2018) is now available. So, we have used the updated GCP data for recalculating the global carbon budgets in the revised manuscript. Since the fossil fuel-derived $CO_2$ emission rates have been slightly downwardly revised, the ocean and land sinks based on the APO data have been slightly decreased. But the changes are at most 0.1 PgC yr$^{-1}$. Consequently, this change has affected very little the conclusion of the original manuscript.

**Reply to minor comments:**

*1) The uncertainties of F and Z_eff should be presented explicitly somewhere in the main text and/or Tables. Did the authors assume the uncertainty of 100% for Z_eff following Keeling and Manning (2014)? Also, for those readers not familiar with O2/N2 studies, it would be better to present the representative values of alpha_F for the period 1998-2016 or the respective values for the periods in Table 2.*

In response to the suggestions about the uncertainties from both Referee #1 and #2, to clarify how we calculated the uncertainties associated with the sink estimations, we have added the following paragraph after the first paragraph in section 3.2.:" The uncertainties in the parameters used for the carbon budget calculation (Eqs. (6) and (7)), which are also listed in Table 2, are briefly discussed here. Note that in this study the estimated uncertainties are ±1σ. Since the ocean outgassing effect is rather speculative, we assumed that the values of $Z_{eff}$ for the individual periods had ±100% uncertainties in accordance with previous studies (e.g. Manning and Keeling, 2006; Tohjima et al., 2008). We adopted uncertainties of ±5% for the fossil fuel-derived $CO_2$ emission rate and ±0.2 PgC yr$^{-1}$ for the atmospheric $CO_2$ increasing rate from Le Quéré, et al. (2018). As for the uncertainties of the observed APO changing rates, we adopted the standard deviations among the sites shown in Fig. 7 (±0.37 per meg yr$^{-1}$ for longer than 10 years and ±0.54 per meg yr$^{-1}$ for 5 years). The estimated uncertainty of the $O_2/N_2$ scale stability (±0.45 per meg yr$^{-1}$) discussed in Section 2. 3, the uncertainty of the $O_2/N_2$ span sensitivity (±3%), and the uncertainty in the global averaged APO associated with the limited atmospheric sampling (±0.2 PgC yr$^{-1}$) discussed in Nevison et al. (2008) were also included in the calculation of the uncertainties in ΔAPO. The uncertainties of $\alpha_B$ and $\alpha_F$ were ±0.10 (Keeling and Manning, 2014) and ±0.04 (Tohjima et al., 2008), respectively. Finally, these uncertainties were propagated to the ocean and land sink uncertainties in accordance with Eqs. (6) and (7)."
Additionally, we have added a column for the values of $\alpha_F$ in Table 2 and a description of the uncertainties in the footnote. Following these changes, Table 2 has been modified as follows:

**Table 2.** Comparison of global carbon budgets based on APO with those from GCP[a,b]

| | Atm. | Fossil | Sink of this study | | Sink of GCP[d] |
|---|---|---|---|---|---|

| Period | ΔAPO [c] | CO$_2$ [d] | fuel [d] | $\alpha_F$ [e] | Z$_{eff}$/1.1 [f] | Ocean | Land | Ocean | Land | Imb. |
|---|---|---|---|---|---|---|---|---|---|---|
| 1998-2016 | -10.3(0.91) | 4.45 | 8.28 | 1.38 | 0.52 | 2.57(0.71) | 1.26(0.89) | 2.24 | 1.46 | 0.13 |
| 2000-2016 | -10.3(0.91) | 4.45 | 8.48 | 1.38 | 0.54 | 2.55(0.73) | 1.48(0.91) | 2.27 | 1.48 | 0.29 |
| 2003-2016 | -9.9(0.91) | 4.58 | 8.83 | 1.38 | 0.52 | 2.35(0.73) | 1.90(0.93) | 2.34 | 1.55 | 0.36 |
| 2000-2004 | -8.8(0.94) | 3.93 | 7.11 | 1.40 | 0.59 | 2.23(0.76) | 0.94(0.90) | 2.01 | 1.30 | −0.14 |
| 2004-2008 | -9.2(0.96) | 4.08 | 8.21 | 1.38 | 0.33 | 1.97(0.62) | 2.17(0.82) | 2.18 | 1.74 | 0.22 |
| 2008-2012 | -10.4(0.98) | 4.19 | 9.05 | 1.37 | 0.54 | 2.54(0.77) | 2.31(0.97) | 2.32 | 1.85 | 0.68 |
| 2012-2016 | -11.6(1.06) | 5.36 | 9.65 | 1.37 | 0.71 | 3.05(0.90) | 1.25(1.09) | 2.55 | 1.26 | 0.49 |

[a]Figures are given in units of per meg yr$^{-1}$ for ΔAPO, mol mol$^{-1}$ for $\alpha_F$, and PgC yr$^{-1}$ for the others.

[b]Figures in parentheses represent the uncertainties.

[c]ΔAPO is based on the data from HAT, COI, and cargo ships (40°S-30°N). The uncertainty in parentheses includes the uncertainty associated with the observations, stability in the O$_2$/N$_2$ scale, uncertainty derived from limited sampling, and uncertainty in the O$_2$/N$_2$ span sensitivity (see text).

[d]These figures are computed from the dataset summarized by the Global Carbon Project (GCP). The uncertainties are ±0.2 PgC yr$^{-1}$ for the atmospheric CO$_2$ and ±5% for the fossil fuel emissions, ±0.5 PgC yr$^{-1}$ for the ocean sinks, and ±0.9PgC yr$^{-1}$ for the land sinks (Le Quéré, et al., 2018).

[e]The uncertainties for $\alpha_F$ are ±0.04 mol mol$^{-1}$ (Tohjima et al., 2008).

[f]The values of Z$_{eff}$ include both global ocean warming and anthropogenic nitrogen deposition effects, and the uncertainties are assumed to be ±100% (see text).

*2) The discussion on the evaluation of the needed interval to suppress the temporal variability in Z_eff is useful in deriving reasonable interannual variations in CO2 sinks from O2/N2 observations. The needed interval was estimated to be 5 years in the paper, and the authors used the ocean heat storage of 0-2000 m layer to estimate Z_eff based on the gas flux / heat flux ratio reported by Keeling and Garcia (2002). However, I think the circulation time of ocean deep layer water is much longer than 5 years. Please explain why the authors have considered the use of heat storage of 0-2000 m to be more reasonable than that of 0-700 m. I suppose there is an implicit assumption in the analysis that the ocean circulation and oxygen concentration are in steady-state from the surface to the deep layer. However, temporal variations found in the 5-years average Zeff in Table2 suggest that the ocean is not in steady-state.*

There are several evidences of an increase in the ocean heat content, suggesting that the present ocean is not in steady-state. The changes in both the heat and O$_2$ contents of the ocean depend on net air-sea exchanges of the heat and O$_2$. Considering the similarity between heat and gas regarding the air-sea exchange, we can assume that the air-sea O$_2$ flux is proportional to the air-sea heat flux (O$_2$ flux/heat flux = constant). Under this assumption, we can evaluate the O$_2$ outgassing flux if we know the total increase in the ocean heat content. In previous studies, ocean heat content data for the 0-700 m layer were used to evaluate the O$_2$ outgassing fluxes. However, Levitus et al. (2012) revealed that a much deeper layer significantly contributed to the ocean heat storage: about one third of the heat storage of the 0-2000 m layer is stored in the 700-2000 m layer. Therefore, we used the data for the 0-2000 m layer in this study following the study of Keeling and Manning (2014). These explanations are given in the fourth paragraph of Section 2.5.

In addition to the gradual ocean warming, there are large inter-annual variations in the air-sea heat exchange, which are attributed to an imbalance of the large seasonality in the air-sea heat fluxes. These inter-annual variations in the air-sea heat fluxes are considered to cause rather large inter-annual variations in the air-sea gas exchanges. In this study, we conclude that the 5-year average would to some extent, but not completely, suppress such interannual variability. These explanations are also given in the fourth paragraph of Introduction.

*3) I think it may be helpful for the reader to note the differences in land and ocean CO2 uptakes expected from the 3% difference in the span sensitivity between NIES and SIO. Has the conclusion about the comparison of the CO2 uptake reported by the present study with those by GCP changed significantly due to the difference in the span sensitivity?*

A 3% higher span sensitivity, which corresponds to the SIO oxygen span sensitivity, results in an approx. 0.3 PgC yr$^{-1}$ increase and decrease in the ocean and land sinks, respectively. These changes, although not negligible, have not changed significantly the conclusion about the comparison of the $CO_2$ sinks between this study and those of GCP. To clearly state the influence of the difference in the span sensitivity, we have added the following sentences after the last sentence of the second paragraph (the third paragraph in the revised manuscript) in section 3.2.: "A 3% higher span sensitivity of the $O_2/N_2$ measurements, which corresponds to the difference in the span sensitivity between SIO and NIES (section 3.1), would result in an increase and decrease of 0.27 PgC yr$^{-1}$ in the ocean and land sinks, respectively, for the three long periods. Although these changes would enlarge the differences in sinks between GCP and this study, the differences are still not significant given the uncertainties of both this study and GCP."

*4) The authors have compared land and ocean CO2 sinks estimated in the present study with those obtained by GCP, with and without the imbalance sinks added. It seems to me that the authors conclude that the differences between the present study and GCP are reduced, both for land and ocean CO2 sinks, by adding the "total" imbalance to the respective sinks. However, I actually think we can only add the imbalance to the land and ocean CO2 sinks based on an appropriate differential distribution. I understand it would be difficult to suggest the best distribution due to uncertainties of the estimated CO2 sinks, but I would like to hear the authors' thoughts on this.*

We cannot draw any certain conclusion about how to partition the budget imbalance between ocean and land sinks because of the large uncertainties associated with the sink estimation of this study, as Referee #1 also pointed out. Nevertheless, the differences in the sinks between the estimates of this study and those of GCP correspond to the best estimation of the partioning of the budget imbalance. Since the budget imbalances seem to increase after 2007 in the pentad averages, we have added a discussion about the partitioning of the imbalances in the revised manuscript. Consequently, we have added one paragraph after the fourth paragraph in section 3.3. to read as: "Shown as discrepancies between the pentad sinks of GCP with and without budget imbalances (Fig. 8), the magnitude of the budget imbalances seems to increase after 2007. For the pentad sinks centered on 2007, 2008 and 2009, the ocean and land sinks of this study agree with those of GCP without and with the budget imbalances, respectively. For the pentad sinks between 2010 and 2014, both the ocean and land sinks of this study are plotted between those of GCP with and without the budget imbalances. Although we cannot show any definitive partitioning of the budget imbalance between ocean and land sinks because of a rather large uncertainty associated with the sink estimations, the above results seem to suggest that the budget imbalances are allocated to land sinks for the former period and to both sinks for the latter period."

*5) P3, line 32: A literal error ": : :heat content,." should be corrected.*

The literal errors have been corrected.

*6) P7, line 19: I think the unit of Z_eff is not TgC yr-1 but PgCyr-1 in this context.*

The unit of $Z_{eff}$ has been corrected to "PgC yr$^{-1}$".

*7) References: Please consolidate the format of references. For example, some journal titles are written in Italic and the others are not.*

We have consolidated the format of references.

*8) Caption in Fig.7: The phrase "changing ratio" should be changed to "changing rate".*

"changing ratio" has been changed to "changing rate".

---

## Author Comment (AC2) · 21 Jun 2019

**Response to review comments on "Global carbon budgets estimated from atmospheric $O_2/N_2$ and $CO_2$ observations in the western Pacific region over a 15-year period"**

**Anonymous Referee #2:**
We would like to thank the anonymous referee for his/her helpful comments and suggestions on our paper. We have revised the manuscript as is described in the following. The referee's comments are in *blue italics*, and the modifications are shown in red.

**Reply to main comments:**

*1) It's unclear how uncertainty in estimating Z_eff comes into play in the carbon budget estimates, especially given its relevance for the shorter timescales considered. The authors do show trends without Z_eff (figure 8) but it's not evident if this is incorporated in the carbon sinks and trends calculation (i.e. unclear if incorporated in grey shading in figure 8 or error estimates in carbon sinks column in Table 2). Perhaps, the authors could evaluate uncertainty in Z_eff from using the upper/lower bounds with and without Z_eff? Further, it's unclear how correcting for Z_eff in the carbon budgets plays out in the 5 year timescales described (as also discussed in Nevison et al. 2008 and elsewhere). It seems, as referred to by the authors, that the ventilation events of 1999-2001 could impact the pentad trends, and thus similar variability during other years probably could have similar effects on other pentad periods (e.g. 2004-2005 dip in pentad ocean sink seems to co-occur with inter-annual variation that don't seem to be fully suppressed?). Finally, an additional and not insignificant component of Z_eff not included in this study is the atmospheric deposition effect, as detailed in Keeling and Manning 2014, which adds about 0.1 (+/-0.1) Pg C/yr, and which should raise Z_eff from 0.1-0.9 Pg C/yr, to 0.2-1.0 Pg C/yr. Overall, I fell the treatment of Z_eff uncertainty within the shorter timescales considered here merits further clarification.*

As both Referee #1 and #2 indicated, the description of the uncertainty in the ocean outgassing effect was unclear in the original manuscript. We assumed a ±100% uncertainty for $Z_{eff}$ for the corresponding budget calculation period. To make it clear, we have added a sentence in the second paragraph in the revised manuscript (see reply (2) for more details): "Since the ocean outgassing effect is rather speculative, we assumed that the values of $Z_{eff}$ for the individual periods had ±100% uncertainties in accordance with previous studies (e.g. Manning and Keeling, 2006; Tohjima et al., 2008)."

As Referee#2 suspected, it seems that the five-year average cannot fully suppress the APO variations associated with the anomalous air-sea gas exchanges. As we discussed in Section 3.1 and as is shown in Fig. 7, the pentad APO trends have a temporal variability of about +1.2 per meg $yr^{-1}$ or +0.5 PgC $yr^{-1}$, which is comparable to the 2004-2005 dip in the pentad ocean sink shown in Fig. 8. To emphasize the limitations of the pentad averaging, we added the following sentences after the first sentence of the second paragraph in Section 3.3: "Nevison et al. (2008) suggested that a decadal or longer period is needed to suppress the influence of the interannual variation in the ocean $O_2$ flux on the carbon sink estimation within ±0.1 PgC $yr^{-1}$ based on an ocean ecosystem model and an atmospheric transport model. In addition, the pentad APO changing rate still contains an uncertainty corresponding to ±0.5 PgC $yr^{-1}$ as is discussed in Section 3.1. Therefore, the anomalous dip in the ocean sink for 2004-2005 might be an error caused by the anomalous ocean $O_2$ flux variations."

In response to the suggestion of Referee #2, we have included the anthropogenic N deposition effect in the $Z_{eff}$ estimation. To explain the anthropogenic nitrogen deposition effect ($Z_{anthN}$), we have separated $Z_{eff}$ into $Z_{anthN}$ and the global ocean warming component ($Z_{gow}$), and added the following paragraph after the fourth paragraph in Section 2.5:

"In addition to the ocean warming effect, Keeling and Manning (2014) introduced recently another ocean outgassing effect caused by atmospheric deposition of excess anthropogenic nitrogen to the open ocean. The excess nitrogen is considered to enhance the ocean biotic production of organic matter, which is associated with the $O_2$ production. Keeling and Manning (2014) evaluated the anthropogenic nitrogen-induced outgassing as being about $0.1 \times 10^{14}$ mol $O_2$ yr$^{-1}$. Since the outgassing effect caused by the anthropogenic nitrogen deposition is small but rather significant, we adopted the effect as $Z_{anthN}$, with a magnitude of 0.12 PgC yr$^{-1}$ (=$0.1 \times 10^{14}$ mol $O_2$ yr$^{-1}$ ×12.01 gC mol$^{-1}$). Eventually, the total outgassing effect, $Z_{eff}$, is expressed as the summation of $Z_{gow}$ and $Z_{anthN}$:

$$Z_{eff} = Z_{gow} + Z_{anthN}. \qquad (9)"$$

*2) I see the need perhaps for a section dedicated to clarifying and detailing the sources, contribution, and methods for calculating carbon budget uncertainty, as it can help clarify the confidence in the pentad trends and conclusions presented here. Table 2, for instance, could incorporate uncertainty due to Z_eff in the carbon sinks column, and in Figure 8 (gray shading). Uncertainty due to undersampling the global signal has also been shown by Nevison et al. (2008) to contribute to uncertainty in estimating budgets on the shorter timescales evaluated here. What is impact of sampling over the western Pacific (_40S-40N) vs. full global sampling on the carbon sink trends? How does uncertainty in alpha_B in using values of 1.1 vs 1.05 affect the uncertainty in carbon sink budgets? Finally, it's unclear how the contribution of measurement uncertainty, due to span calibration of the gas chromatographic technique and potential longterm drift shared across all cylinders, is incorporated in the uncertainty analysis.*

In the original manuscript, we didn't incorporate the anthropogenic N deposition effect on $Z_{eff}$, the uncertainty associated with the global average of APO from limited samples, uncertainty due to span sensitivity of the gas chromatographic technique, and the potential long-term drift among the $O_2/N_2$ reference cylinders. In addition, the uncertainty of ±0.05 for $\alpha_B$ in the original manuscript should be increased to ±0.10. Considering these uncertainties, we have carefully reevaluated the uncertainties. To make it clear, we have modified the original manuscript as follows:

To clarify the long-term stability of the $O_2/N_2$ scale, we have added the following sentences at the end of Section 2.3: "However, this stability test cannot exclude the possibility that the $O_2/N_2$ ratios of the reference gases drift across all the cylinders rather uniformly. There are several mechanisms that affect the $O_2/N_2$ ratios of the gases within the high-pressure cylinders, including corrosion of the inner surface, leakage, thermal diffusion and gravitational fractionation. Keeling et al. (2007) assessed carefully and comprehensively the influences of those potential mechanisms on the long-term stability of the $O_2/N_2$ ratio of the reference gases and obtained an estimated uncertainty of ±0.4 per meg yr$^{-1}$. We also treated the reference cylinders which were kept horizontally in a thermally insulated box, with the greatest care (Tohjima et al., 2008). Therefore, we adopted the value of ±0.4 per meg yr$^{-1}$ as the long-term drift of the reference gases caused by the above degradation effects. Consequently, we assumed that the total uncertainty of the long-term stability of the $O_2/N_2$ reference scale was ±0.45 per meg yr$^{-1}$ (=$(0.2^2+0.4^2)^{1/2}$) in this study."

As for the uncertainty of $\alpha_B$, we have added the following sentences at the end of Section 2.4: "Considering the recent reports about the global net $-O_2/CO_2$ exchange ratio, Keeling and Manning (2014) revised the uncertainty of $\alpha_B$ upward from ±0.05 (Severinghaus, 1995) to ±0.10. Thus, we also adopted ±0.10 for the uncertainty of $\alpha_B$ in this study."

Finally, to clarify how we computed the total uncertainties associated with the global sink estimations, we have added the following paragraph after the first paragraph in Section 3.2:

"The uncertainties in the parameters used for the carbon budget calculation (Eqs. (6) and (7)), which are also listed in Table 2, are briefly discussed here. Note that in this study the estimated uncertainties are $\pm 1\sigma$. Since the ocean outgassing effect is rather speculative, we assumed that the values of $Z_{eff}$ for the individual periods had $\pm 100\%$ uncertainties in accordance with previous studies (e.g. Manning and Keeling, 2006; Tohjima et al., 2008). We adopted uncertainties of $\pm 5\%$ for the fossil fuel-derived $CO_2$ emission rate and $\pm 0.2$ PgC yr$^{-1}$ for the atmospheric $CO_2$ increasing rate from Le Quéré, et al. (2018). As for the uncertainties of the observed APO changing rates, we adopted the standard deviations among the sites shown in Fig. 7 ($\pm 0.37$ per meg yr$^{-1}$ for longer than 10 years and $\pm 0.54$ per meg yr$^{-1}$ for 5 years). The estimated uncertainty of the $O_2/N_2$ scale stability ($\pm 0.45$ per meg yr$^{-1}$) discussed in Section 2. 3, the uncertainty of the $O_2/N_2$ span sensitivity ($\pm 3\%$), and the uncertainty in the global averaged APO associated with the limited atmospheric sampling ($\pm 0.2$ PgC yr$^{-1}$) discussed in Nevison et al. (2008) were also included in the calculation of the uncertainties in $\Delta$APO. The uncertainties of $\alpha_B$ and $\alpha_F$ were $\pm 0.10$ (Keeling and Manning, 2014) and $\pm 0.04$ (Tohjima et al., 2008), respectively. Finally, these uncertainties were propagated to the ocean and land sink uncertainties in accordance with Eqs. (6) and (7)."

In accordance with the uncertainty revision, we have also revised the uncertainties of the $O_2/N_2$ and APO changing rates listed in Table 1. The uncertainties of $\pm 0.2$ or $\pm 0.3$ per meg yr$^{-1}$ have increased to $\pm 0.8$ per meg yr$^{-1}$. In addition, we have also redrawn Fig. 8 as follows:

[Figure]

**Fig. 8.** Temporal variations in (a) ocean and (b) land biospheric sinks estimated from APO variations of this study (red) and process-based models of GCP (blue). The thin broken lines represent the annual sinks and the thick lines represent the pentad sinks. The purple lines represent the pentad sinks based on APO without ocean outgassing correction ($Z_{eff}$) and the light blue lines represent the sinks of GCP with the imbalance sinks added. The uncertainty associated with the pentad sinks with $Z_{eff}$ corrections are shown as shaded area.

*3) The comparison against the GCP could be elaborated on a bit more, as it raises important issues in the field. The authors could elaborate further (through existing or new figure/table) how different estimates reported by GCP compare to the APO method, including hindcast ocean models and ocean observation based*

*products, all of which are readily available in the GCP product as globally integrated fluxes:*
*https://www.icoscp.eu/GCP/2018. It is interesting that the comparison to the GCP mean showcases*
*similarities in magnitude and in temporal evolution of pentads. The point that the uptake of carbon by the*
*ocean is larger than expected from atmospheric increase alone*
*is very interesting. How do the decadal trends (2000-2016) in this study compare to the pCO2 based air-sea*
*flux timeseries by Landshutzer et al (2016) and Rodenbeck et al (2013), as both of these estimates seem to*
*show larger decadal variability than the ocean models? These items may be beyond the current scope of this*
*study, but could substantially improve the impact of this paper with (hopefully?) relatively minor figure/text*
*additions.*

We have carefully compared our pentad ocean sinks based on APO with those of GCP and the $pCO_2$-based estimations and found that the increasing trend of the ocean sinks based on APO was close to those based on the $pCO_2$ observations. To explain this clearly, we have added the following sentences after the second to the last sentence of the second paragraph in Section 3.3: "For a detailed comparison, the global ocean sinks based on $pCO_2$ observations and interpolation techniques (Landschützer et al., 2016; Rödenbeck et al., 2014) for the period of 1990-2017 are plotted in Fig. 9 together with the ocean sinks of this study and GCP. Note that the extended $pCO_2$-derived ocean sinks were given as supplementary data of Le Quéré, et al. (2018) and those sinks were uniformly inflated by 0.78 PgC $yr^{-1}$ to compensate for the pre-industrial steady state source of $CO_2$ derived from riverine input of carbon to the ocean (Resplandy et al., 2018). As you can see, both the GCP and $pCO_2$-derived ocean sinks show changes in the trends between before and after 2001 while the magnitude of the changes in the $pCO_2$-derived sinks are larger. The increasing rates determined by a linear regression during 2001-2014 are 0.08 ± 0.01 PgC $yr^{-2}$ in Landschüzer et al. (2016) and 0.07 ± 0.02 PgC $yr^{-2}$ in Rödenbeck et al. (2014), which are more consistent with the rate found in this study. Therefore, our result seems to support a previous conclusion that the recent increase in the ocean sinks exceeds the increasing trend of ocean sink expected only from the atmospheric $CO_2$ increase (Landschützer et al., 2015; DeVries et al., 2017)." In accordance of this change, we have modified the third sentence of Conclusion 3) as "The pentad ocean sinks showed an overall increasing trend for the entire period (2001-2014) with a linear increasing rate of 0.08 ± 0.02 PgC $yr^{-2}$. This increasing rate was about two times larger than that for the GCP ocean sinks (0.04 ± 0.01 PgC $yr^{-2}$) but was consistent with those for the global ocean sinks based on $pCO_2$ observations and interpolation techniques (Landschützer et al., 2016; Rödenbeck et al., 2014)." We have also added Resplandy et al. (2018) to Reference and Fig. 9 showing ocean sinks based on the APO data, process-based models (GCP), and $pCO_2$ observations as follows:

[Figure]

**Fig. 9.** Comparison of the temporal variations of the ocean sinks based on the APO data of this study (red), global ocean biogeochemistry models (GOBMs) of GCP (blue), and $pCO_2$ data of Landschützer et al. (2016) (light blue) and Rödenbeck et al. (2014) (orange). The broken lines represent the regression lines for the corresponding data during 2001-2014. Note that the $pCO_2$-based ocean sinks are adjusted for the pre-industrial ocean $CO_2$ emissions ($\pm 0.78$ PgC yr$^{-1}$) caused by riverine $CO_2$ input to the ocean (Resplandy et al., 2018).

In addition to the above modifications, we have changed the GCP-reported data (fossil fuel emissions, atmospheric accumulation, and global sinks) from Global Carbon Budget 2017 to the data from Global Carbon Budget 2018 (Le Quéré, et al., 2018). We have used the updated GCP data for recalculating the global carbon budgets in the revised manuscript. Because the fossil fuel-derived $CO_2$ emission rates have been slightly downwardly revised, the ocean and land sinks based on the APO data have been slightly decreased. But the changes are at most 0.1 PgC yr$^{-1}$. Consequently, this change has affected very little the conclusion of the original manuscript.

**Reply to minor issues:**

*Pg2 L27: "The estimated value for _F is about 1.10±0.05 (Severinghaus, 1995) and that for _B is about 1.4 (Keeling, 1988)." Should be the other way around: _B is 1.10 and _F is 1.4.*

"The estimated value for $\alpha_F$ is about 1.10±0.05 (Severinghaus, 1995) and that for $\alpha_B$ is about 1.4 (Keeling, 1988)" has been changed to "The estimated value for $\alpha_B$ is about 1.1 (Severinghaus, 1995) and that for $\alpha_F$ is about 1.4 (Keeling, 1988)."

*Perhaps add citations for Equations (1), (2), and (3)?*

Citation for Eq. (1), (2), and (3): We have added the citation, Manning and Keeling (2006), for these equations.

*P3 L1, this paragraph could use a brief explanation of APO concept as a tracer for those not familiar with APO, i.e. cancellation of terrestrial influence, etc*

In accordance with the suggestion, we have added the following sentence after the first sentence of the paragraph: "Since the APO is defined to be invariant with respect to the land biotic exchange, the secular trend in the APO is determined by fossil fuel combustions which cause a gradually decreasing trend in APO, and the air-sea gas exchange."

*Pg 7 L29, shouldn't Z_eff be in PgC/yr?*

The unit "TgC yr$^{-1}$" has been altered to "PgC yr$^{-1}$".

*Pg 9 Line 20, the ENSO topic deserves a bit more clarification here. It would be good to preface the ENSO sentence with the findings of Rodenbeck et al 2008, who suggest anomalous outgassing of APO during El Niño, while Tohjima et al 2015 show a suppressed peak instead, and clarify that Eddebbar et al (2017) reconcile this apparent discrepancy through a model-simulated zonal dipole-like ENSO response in the equatorial Pacific, and that enhanced observational zonal coverage in this region is needed to constrain the full basin ENSO response.*

In response to the Referee's suggestion, we have modified the relevant part as "Conducting atmospheric inversion analyses based on the APO data from the Scripps observation network,

Rödenbeck et al. (2008) suggested anomalous outgassing of APO from the equatorial region during El Niño periods, while Tohjima et al. (2015) found a suppressed equatorial peak during El Niño periods based on the western Pacific observations. Eddebbar et al. (2017) reconciled these conflicting results by predicting the existence of a zonal dipole-like ENSO response in the equatorial Pacific based on several ocean process-based models and an atmospheric transport model. These results suggest that an enhanced zonal coverage of the atmospheric observations in the equatorial Pacific is needed to constrain the full basin-scale ENSO response. We can see a considerable suppression of the equatorial peak during the strong 2015/2016 El Niño event in Fig. 6c, which was not reported in Tohjima et al. (2015). Any detailed discussion about the temporal variation of the equatorial peak during the 2015/2016 El Niño event is, however, beyond the scope of this study and will be given elsewhere."

**Suggested editing notes:**
*Pg 2 Line 5: remove "still", and add year by which emissions rose to 10 Pg C/yr?*

"…the global fossil fuel-derived $CO_2$ emissions in recent years still increased gradually and rose toward 10 PgC yr$^{-1}$ (Boden et al. 2017)" has been changed to "…the global fossil fuel-derived $CO_2$ emissions in recent years still increased gradually and rose to 9.9 PgC yr$^{-1}$ by 2014 (Boden et al. 2017)".

*Pg 2 Line 6: "Paris Agreement . . . aimed to balance the anthropogenic greenhouse gas emissions and natural removals in the second half of this century. . .", I suggest editing to: " . . . aimed to reduce anthropogenic greenhouse gas emissions to maintain the increase in global mean surface temperature well below 2_C by 2100, . . ."?*

The relevant part has been modified to "…, the Paris Agreement adopted at COP21 in 2015 aimed to reduce the anthropogenic greenhouse gas emissions to maintain the increase in global mean surface temperatures well below 2°C by 2100, …".

*Pg3 L8, suggest deleting "In these days".*

"In these days" has been deleted.

*Pg3 L21, "which reduces the ventilation of the seawater.", suggest instead: "which reduces the ventilation of interior water masses."*

We have changed the relevant sentence from "the ventilation of the seawater" to "the ventilation of interior water masses".

*Pg3 L26: replace "huge" with "large"*

"huge" has been replaced with "large".

*Pg 14 L 20: Not sure I understand this sentence: "This means that the changing trends of carbon budgets may be evaluated by the at least decadal APO data." Suggest rephrasing and/or elaborating further?*

The ambiguous fourth conclusion has been deleted. We have also modified the first two sentences of the last paragraph in Section 3.3, "From the above discussions, we feel …in the temporal resolution.", to "From the above discussions, we feel that a five-year duration effectively suppresses to some extent the anomalous variations in the carbon budget estimations

based on APO, which are considered to be caused by the imbalance of the seasonal air-sea $O_2$ exchange. Probably, the five-year average suppresses the variability of $Z_{eff}$ to a level of $\pm 0.5$ PgC yr$^{-1}$ as is discussed in Section 3.1."

*Pg 12 L 32. Replace "stagnant" with "stagnancy"*

"stagnant" has been replaced with "stagnancy".

*Pg13 L 1: replace "in spite of" with "despite"*

"in spite of" has been replaced with "despite".

---

## Author Response (AR2)

*Comments to the Author:*

*Thank you for the work that you have undertaken to address the review comments. You have obviously done this carefully and I'm sure the manuscript has been improved as a result. I noticed that a few minor technical corrections are needed.*

*p10, line 5: 'ingassing' instead of 'ingassig'*

*p14, line 4: Suggest delete 'As you can see'*

*p14, line 16: Suggest 'stagnation' instead of 'stagnancy'*

*Fig 5. There are numbers on the y axis, 360, 0, 0. It is not clear which of the timeseries they relate to so perhaps they should be removed. Perhaps the figure caption needs to say that the timeseries have been offset to allow them to be plotted on the same panel.*

*Fig 7 caption: I think there are 3 more places where 'ratio' needs to be changed to 'rate'.*

All above corrections and suggestions have been included in the final manuscript.

As for Fig. 5, the numbers of the y-axis represent the values for HAT data. To clarify it, we have redrawn Fig. 5. As the co-editor suggested, the time-series have been offset to allow them to be plotted on the same panels. Consequently, we have changed the figure caption to "**Fig. 5.** Time series of the atmospheric $CO_2$ mole fraction (left), $O_2/N_2$ ratio (middle), and APO (right) of the flask samples obtained from the NIES flask sampling network shown in **Fig. 1**. Observed data from COI, HAT, and cargo ships operating between 40ºS and 30ºN were used for the global carbon budget calculation. The time series of $CO_2$, $O_2/N_2$ and APO are offset by 20 ppm, 150 per meg, and 100 per meg, respectively, to allow them to be plotted on the same panels. The numbers on the y-axis represent the values for the data at HAT."